# PROCEEDINGS A

computer modelling and simulation, statistics, biophysics

statistical learning theory, sparse regression, differential equations, stability selection, PAR proteins, machine learning

**Author for correspondence:**
Ivo F. Sbalzarini
e-mail: sbalzarini@mpi-cbg.de

# Stability selection enables robust learning of differential equations from limited noisy data

Suryanarayana Maddu[1,2,3,4],
Bevan L. Cheeseman[1,2,3,4], Ivo F. Sbalzarini[1,2,3,4] and
Christian L. Müller[5]

[1]Faculty of Computer Science, Technische Universität Dresden, Dresden, Germany
[2]Max Planck Institute of Molecular Cell Biology and Genetics, Dresden, Germany
[3]Center for Systems Biology Dresden, Dresden, Germany
[4]Cluster of Excellence Physics of Life, TU Dresden, Germany
[5]Center for Computational Mathematics, Flatiron Institute, New York, NY, USA

SM, 0000-0003-2702-4427; IFS, 0000-0003-4414-4340;
CLM, 0000-0002-3821-7083

We present a statistical learning framework for robust identification of differential equations from noisy spatio-temporal data. We address two issues that have so far limited the application of such methods, namely their robustness against noise and the need for manual parameter tuning, by proposing stability-based model selection to determine the level of regularization required for reproducible inference. This avoids manual parameter tuning and improves robustness against noise in the data. Our stability selection approach, termed PDE-STRIDE, can be combined with any sparsity-promoting regression method and provides an interpretable criterion for model component importance. We show that the particular combination of stability selection with the iterative hard-thresholding algorithm from compressed sensing provides a fast and robust framework for equation inference that outperforms previous approaches with respect to accuracy, amount of data required, and robustness. We illustrate

the performance of PDE-STRIDE on a range of simulated benchmark problems, and we demonstrate the applicability of PDE-STRIDE on real-world data by considering purely data-driven inference of the protein interaction network for embryonic polarization in *Caenorhabditis elegans*. Using fluorescence microscopy images of *C. elegans* zygotes as input data, PDE-STRIDE is able to learn the molecular interactions of the proteins.

## 1. Introduction

Predictive mathematical models, validated in experiments, are of key importance for the scientific understanding of natural phenomena. While this approach has been particularly successful in describing spatio-temporal dynamical systems in physics and engineering, it has not seen the same degree of success in other scientific fields, such as neuroscience, biology, finance, and ecology. This is because the underlying first principles in these areas remain largely elusive. Nevertheless, modelling in those areas has seen increasing use and relevance to help formulate simplified mathematical equations where sufficient observational data are available for validation [1–5]. In biology, modern high-throughput technologies have enabled collection of large-scale datasets, ranging from genomics, proteomics and metabolomics data, to microscopy images and videos of cells and tissues. These datasets are routinely used to infer parameters in hypothesized models, or to perform model selection among a finite number of alternative hypotheses [6–8]. The amount and quality of biological data, as well as the performance of computing hardware and computational methods, have now reached a level that promises direct inference of mathematical models of biological processes from the available experimental data. Such data-driven approaches seem particularly valuable in cell and developmental biology, where first principles are hard to come by, but large-scale imaging data are available, along with an accepted consensus of which phenomena a model could possibly entail. In such scenarios, data-driven modelling approaches have the potential to uncover the unknown first principles underlying the observed biological dynamics.

Biological dynamics can be formalized at different scales, from discrete molecular processes to the continuum mechanics of tissues. Here, we consider the macroscopic, continuum scale where spatio-temporal dynamics are modelled by partial differential equations (PDEs) over coarse-grained state variables [3,9]. PDE models have been used to successfully address a range of biological problems from embryo patterning [10] to modelling gene-expression networks [11,12] to predictive models of cell and tissue mechanics during growth and development [13,14]. They have shown their potential to recapitulate experimental observables in cases where the underlying physical phenomena are known or have been postulated [15–17]. In many biological systems, however, the governing PDEs are not (yet) known, which slows progress in discovering the underlying physical principles. Thus, it is desirable to verify existing models and discover new ones by extracting governing laws directly from measured spatio-temporal data.

For given observable spatio-temporal dynamics, with no governing PDE known, several proposals have been put forward to learn mathematically and physically interpretable PDE models. The earliest work in this direction [18] frames the problem of 'PDE learning' as a multivariate nonlinear regression problem where each component in the design matrix consists of space and time differential operators and low-order nonlinearities computed directly from data. Then, the alternating conditional expectation (ACE) algorithm [19] is used to compute both optimal element-wise nonlinear transformations of each component and their associated coefficients. In [20], the problem is formulated as a linear regression problem with a fixed predefined set of space and time differential operators and polynomial transformations that are computed directly from data. Then, backward elimination is used to identify a compact set of PDE components by minimizing the least-square error of the full model and pruning terms

that worsen the fit the least. In the statistics literature [21,22], the PDE learning problem has been formulated as a Bayesian estimation problem where the observed dynamics are learned via non-parametric approximation, and a PDE representation serves as the prior to compute the posterior estimates of the PDE coefficients. Recent influential work revived the idea of jointly learning the structure *and* the coefficients of PDE models from data in discrete space and time using sparse regression [23–25]. Approaches such as SINDy [23] and PDE-FIND [24] compute a large pre-assembled dictionary of possible PDE terms from data and identify the most promising components by penalized linear regression. PDE-FIND was able to learn different types of PDEs from simulated spatio-temporal data, including Burgers, Kuramoto-Sivashinsky, reaction-diffusion, and Navier–Stokes equations. PDE-FIND's performance was evaluated on noise-free simulated data as well as data with up to 1% additive noise and showed a critical dependence on proper tuning of the regularization parameters, which are typically unknown in practice. Recent works attempted to alleviate this dependence by using Bayesian sparse regression for model uncertainty quantification [26] or information criteria for parameter tuning [27]. There is also a growing body of literature that considers deep neural networks for PDE learning [28–30]. For instance, the deep feed-forward network PDE-NET [28] has been shown to directly learn computable, discretized forms of the underlying governing PDEs for forecasting [28,31]. PDE-NET exploits the connection between differential operators and order-of-sum rules of convolution filters [32] to constrain network layers to learning valid discretized differential operators. The forecasting capability of this approach was numerically demonstrated for predefined linear differential operator templates. A compact and interpretable symbolic identification of the PDE structure, however, is not available with this approach.

Here, we ask the question whether and how it is possible to extend the class of sparse regression inference methods to work on limited amounts of noisy experimental data. We present a statistical learning framework, PDE-STRIDE (STability-based Robust IDEntification of PDEs), to robustly infer PDE models from noisy spatio-temporal data without requiring manual tuning of learning parameters, such as regularization constants. PDE-STRIDE is based on the statistical principle of stability selection [33,34], which provides an interpretable criterion for any term's inclusion in the learned PDE in a data-driven manner. Stability selection can be used with any sparsity-promoting regression method, including LASSO [33,35], iterative hard thresholding (IHT) [36], hard thresholding pursuit (HTP) [37] or sequential thresholding ridge regression (STRidge) [24]. PDE-STRIDE therefore provides a drop-in solution to rendering existing inference tools more robust, while reducing the need for parameter tuning. In our benchmarks, the combination of stability selection with de-biased iterative hard thresholding (IHT-d) empirically shows the best performance and highest consistency w.r.t. perturbations of the dictionary matrix and the sampling of the data.

This paper is organized as follows: §2 provides the mathematical formulation of the sparse regression problem and discusses how the design matrix is assembled. We also review the concepts of regularization paths and stability selection and discuss how they are combined in the proposed method. The numerical results in §3 highlight the performance and robustness of PDE-STRIDE for recovering different PDEs from noise-corrupted data. We also perform achievability analysis of PDE-STRIDE + IHT-d for consistency and convergence of the recovery probabilities with increasing sample size. Section 4 demonstrates that the robustness of the proposed method is sufficient for real-world applications. We consider learning a protein interaction model from noisy biological microscopy images of membrane protein dynamics in a *Caenorhabditis elegans* zygote. Section 5 provides a summary of our results and highlights future challenges for data-driven PDE learning.

## 2. Problem formulation and optimization

We outline the problem formulation underlying the data-driven PDE inference considered here. We review important sparse regression techniques and introduce the concept of stability selection used in PDE-STRIDE.

## (a) Problem formulation for partial differential equations learning

We propose a framework for stable estimation of the structure and parameters of the governing equations of continuous dynamical systems from discrete spatio-temporal measurements or observations. Specifically, we consider PDEs for the multidimensional state variable $u \in \mathbb{R}^d$ of the form shown in equation (2.1), composed of polynomial nonlinearities (e.g. $u^2, u^3$), spatial derivatives (e.g., $u_x, u_{xx}$) and the parametric dependence modelled through $\varXi \in \mathbb{R}^p$.

$$\frac{\partial u}{\partial t} = \mathcal{F}([u, u^2, u^3, u_{xx}, uu_x, \ldots], x, t, \varXi). \tag{2.1}$$

Here, $\mathcal{F}(\cdot)$ is the function map that models the spatio-temporal nonlinear dynamics of the system. We limit ourselves to forms of the function map $\mathcal{F}(\cdot)$ that can be written as linear combinations of polynomial nonlinearities, spatial derivatives and combinations of both. For instance, for a one-dimensional ($d = 1$) state variable $u$, the function map can take the form

$$\frac{\partial u}{\partial t} = \underbrace{\xi_0 + \xi_1 u + \xi_2 \frac{\partial u}{\partial x} + \xi_3 u \frac{\partial u}{\partial x} + \xi_4 u^2 + \cdots + \xi_k u^3 \frac{\partial^2 u}{\partial x^2} + \cdots}_{\mathcal{F}(\cdot)}, \tag{2.2}$$

where $\xi_k$ are the coefficients of the PDE components for $k \geq 0$. The continuous PDE of the form described in equation (2.2), with appropriate coefficients $\xi_k$, holds true for all continuous space and time points $(x, t)$ in the domain of the model. Numerical solutions of the PDE try to satisfy the equality relation in equation (2.2) for reconstituting the nonlinear dynamics of a dynamical system at discrete space and time points $(x_m, t_n)$. We assume that we have access to $N$ noisy observational data points $\tilde{u}_{m,n}$ of the state variable $u$ at such discrete space and time points. The measurement errors are assumed to be independent and identically distributed following a normal distribution with mean zero and variance $\sigma^2$.

We follow earlier works [20,24,25] and construct a large dictionary of potential PDE components using discrete approximations of the terms from the data $\tilde{u}_{m,n}$. For instance, for the one-dimensional example in equation (2.2), the discrete approximation of $p$ PDE terms can be written in vectorized form as a linear regression problem

$$\underbrace{\begin{bmatrix} | \\ u_t \\ | \end{bmatrix}}_{U_t} = \underbrace{\begin{bmatrix} | & | & | & | & | & | \\ 1 & u & u_x & \ldots & u^3 u_{xx} & \ldots \\ | & | & | & | & | & | \end{bmatrix}}_{\varTheta} \xi. \tag{2.3}$$

Here, the left-hand side vector $U_t \in \mathbb{R}^N$ contains the discrete approximations of the time derivatives of $u$ at the data points and represents the response or outcome vector of the linear regression. Each column of the dictionary or design matrix $\varTheta \in \mathbb{R}^{N \times p}$ represents the discrete approximation of one PDE component, i.e. one of the terms in equation (2.2), at the $N$ discretization points in space and time. Each column is interpreted as a potential predictor of the response vector $U_t$. The vector $\xi = [\xi_0, \xi_1, \ldots \xi_{p-1}]^\top \in \mathbb{R}^p$ is the vector of unknown PDE coefficients, i.e. the pre-factors of the terms in equation (2.2).

Both $U_t$ and $\varTheta$ need to be constructed from numerical approximations of the temporal and spatial derivatives of the observed state variables. There is a rich literature in numerical analysis on this topic (e.g. [38,39]). Here, we approximate the time derivatives by first-order forward finite differences from $\tilde{u}$ (i.e. the explicit Euler scheme) after initial denoising of the data. Similarly, the spatial derivatives are computed by second-order central finite differences. For denoising, we use truncated singular value decomposition (SVD) with a cut-off at the elbow of the singular values curve, as shown in the electronic supplementary material, figures S1 and S2.

Given the general linear regression ansatz in equation (2.3), we formulate the data-driven PDE inference problem as a regularized optimization problem of the form

$$\hat{\xi}^\lambda = \arg\min_{\xi} \left( h(\xi) + \lambda g(\xi) \right), \tag{2.4}$$

where $\hat{\xi}^\lambda$ is the minimizer of the objective function, $h(\cdot)$ is a smooth convex data-fitting function, $g(\cdot)$ a regularization or penalty function and $\lambda \geq 0$ is a scalar regularization parameter that balances data fitting and regularization. The function $g(\cdot)$ is not necessarily convex or differentiable. We follow previous works [20,24,25] and consider the standard least-squares data-fitting term

$$h(\xi) = \frac{1}{2}||U_t - \Theta\xi||_2^2. \tag{2.5}$$

The choice of the penalty function $g(\cdot)$ influences the properties of the coefficient estimates $\hat{\xi}^\lambda$. We seek to identify a small subset of PDE components among the $p$ possible ones that accurately predict the time evolution of the state variables [23–25]. This implies that we want to identify a sparse coefficient vector $\hat{\xi}^\lambda$, thus resulting in an *simple and interpretable* PDE model. This can be achieved through sparsity-promoting penalty functions $g(\cdot)$. We next consider different choices for $g(\cdot)$ that enforce sparsity in the coefficient vector and review algorithms that solve the associated optimization problems.

## (b) Sparse optimization for partial differential equations learning

The least-squares loss in equation (2.5) can be combined with different sparsity-promoting penalty functions $g(\cdot)$. The prototypical example is the $\ell_1$-norm $g(\cdot) = || \cdot ||_1$ leading to the LASSO formulation of sparse linear regression [35]:

$$\hat{\xi}^\lambda = \arg\min_\xi \left( \underbrace{\frac{1}{2}||U_t - \Theta\xi||_2^2}_{h(\cdot)} + \underbrace{\lambda||\xi||_1}_{g(\cdot)} \right). \tag{2.6}$$

The LASSO objective comprises a convex smooth loss and a convex non-smooth regularizer. For this class of problems, efficient optimization algorithms exist that can exploit the properties of the functions and come with convergence guarantees. Important examples include coordinate-descent algorithms [40,41] and proximal algorithms, including the Douglas–Rachford algorithm [42] and the projected (or proximal) gradient method, also known as the forward-backward algorithm [43]. In signal processing, the latter schemes are also known as iterative shrinkage-thresholding algorithms (ISTA, see [44] and references therein), which can be extended to non-convex penalties. Although LASSO has been previously used for PDE learning [25], the statistical performance of LASSO estimates is known to deteriorate if certain conditions on the design matrix are not met. For example, the studies in [45,46] provide sufficient and necessary conditions, called the *irrepresentable conditions*, for consistent variable selection using LASSO, essentially excluding strong correlations of the predictors in the design matrix. These conditions are, however, difficult to check in practice, as they require knowledge of the true components of the model. One way to relax these conditions is via randomization. The randomized LASSO [33] therefore considers the objective

$$\hat{\xi}^\lambda = \arg\min_\xi \left( \underbrace{\frac{1}{2}||U_t - \Theta\xi||_2^2}_{h(\cdot)} + \underbrace{\lambda \sum_{k=1}^p \frac{|\xi_k|}{W_k}}_{g(\cdot)} \right), \tag{2.7}$$

where each $W_k$ is an *i.i.d.* random variable uniformly distributed over $[\alpha, 1]$ with $\alpha \in (0, 1]$. For $\alpha = 1$, randomized LASSO reduces to standard LASSO. Randomized LASSO has been shown to successfully overcome the limitations of LASSO in handling correlated components in the dictionary [33], while simultaneously preserving the overall convexity of the objective function. As part of our PDE-STRIDE framework, we evaluate the performance of randomized LASSO in the context of PDE learning using cyclical coordinate descent [41].

The sparsity-promoting property of the (weighted) $\ell_1$-norm comes at the expense of considerable bias in the estimation of the non-zero coefficients [46], thus leading to reduced variable selection performance in practice. This drawback can be alleviated by using non-convex

penalty functions [47,48], allowing near-optimal variable selection performance at the cost of needing to solve a non-convex optimization problem. For instance, using the $\ell_0$-'norm' (which counts the number of non-zero elements of a vector) as regularizer $g(\cdot) = || \cdot ||_0$ leads to the NP-hard problem

$$\hat{\xi}^\lambda = \arg \min_\xi \left( \underbrace{\frac{1}{2}||U_t - \Theta\xi||_2^2}_{h(\cdot)} + \underbrace{\lambda||\xi||_0}_{g(\cdot)} \right). \tag{2.8}$$

This formulation has found widespread applications in compressed sensing and signal processing. Algorithms that deliver approximate solutions to equation (2.8) include greedy optimization strategies like orthogonal matching pursuit [49,50], compressed sampling matching pursuit (CoSaMP) [51] and subspace pursuit [52]. We here consider the IHT algorithm [36,53], which belongs to the class of ISTA algorithms. Given the design matrix $\Theta$ and the measurement vector $U_t$, IHT computes sparse solutions $\hat{\xi}$ by applying a nonlinear shrinkage (thresholding) operator to gradient descent steps in an iterative manner. One step in the iterative scheme reads

$$\xi^{n+1} = H_\lambda(\xi^n + \Theta^\top(U_t - \Theta\xi^n)) = T\xi^n, \quad H_\lambda(x) = \begin{cases} 0 & x \le \sqrt{\lambda}, \\ x & \text{otherwise.} \end{cases} \tag{2.9}$$

The operator $H_\lambda(x)$ is the nonlinear hard-thresholding operator. Convergence of the above iteration is guaranteed if and only if $||U_t - \Theta\xi^{n+1}||_2^2 < (1 - c)||U_t - \Theta\xi^n||_2^2$ in each iteration for some constant $0 < c < 1$. Under the condition that $||\Theta||_2 < 1$, the IHT algorithm is guaranteed to not increase the cost function in equation (2.8) (Lemma 1 in [36]). The IHT algorithm can be viewed as a thresholded version of the classic Landweber iteration [54]. The fixed points $\xi^*$ for which $\xi^* = T\xi^*$ for the nonlinear operator $T$ in equation (2.9) are local minima of the cost function in equation (2.8) (Lemma 3 in [36]). Under the same condition on the design matrix, i.e. $||\Theta||_2 < 1$, the optimal solution of the cost function in equation (2.8) thus belongs to the set of fixed points of the IHT algorithm (theorem 2 in [36] and theorem 12 in [55]). Although the IHT algorithm comes with theoretical convergence guarantees, the resulting fixed points are not necessarily sparse [36].

Here, we propose a modification of the IHT algorithm that will prove to be particularly well suited for solving PDE learning problems. Following a proposal in [37] for the HTP algorithm, we equip the IHT algorithm with an additional debiasing step. This involves solving at each iteration a least-squares problem restricted to the support $S^{n+1} = \{k : \xi_k^{n+1} \ne 0\}$ obtained from the $n$-th IHT iteration. We refer to this form of IHT as *iterative hard thresholding with debiasing* (IHT-d). In this two-step algorithm, the standard IHT step serves to extract the explanatory variables, while the debiasing step approximately debiases (or re-fits) the coefficients restricted to the currently active support [56]. Rather than solving the least-squares problem to optimality, we use gradient descent steps until a loose upper bound on the least-squares re-fit is satisfied, $||U_t - \Theta\xi^{n+1}||_2^2 \le \lambda|S^{n+1}|$. This prevents over-fitting by attributing low confidence to large supports, which reduces computational overhead and renders the algorithm practical. The complete IHT-d procedure is detailed in Algorithm 1 in the electronic supplementary material. In PDE-STRIDE, we compare IHT-d with a heuristic iterative algorithm, Sequential Thresholding of Ridge regression (STRidge), that also uses $\ell_0$ penalization and is available in PDE-FIND [24].

## (c) Stability selection

The practical performance of sparse optimization techniques in PDE learning hinges on proper selection of the regularization parameter $\lambda$ that balances model fit and model complexity. In model discovery tasks on real experimental data, a wrong choice of the regularization parameter could result in incorrect PDE model selection even if true model discovery would have been, in principle, achievable. In statistics, a large number of tuning parameter selection criteria are available, ranging from cross-validation approaches [57] to information criteria [58], or formulations that allow joint learning of model coefficients and tuning parameters [59,60]. Here, we advocate stability-based model selection [33] for robust PDE learning. The statistical principle

of stability [61] has been put forward as one of the pillars of modern data science and statistics. It provides an intuitive approach to model selection [33,34,62]. Stability selection has found widespread application from the analysis of gene regulatory networks [63] to graphical models [64] and ecological studies [65] .

In the context of sparse regression, stability selection [33] proceeds as follows (see also figure 1 for an illustration): given a design matrix $\Theta$ and measurement vector $U_t$, generate random subsample indices $I_i^* \subset \{1, \ldots, N\}$, $i = 1, \ldots, B$ of equal size $|I_i^*| = N/2$ and produce reduced sub-designs $\Theta[I_i^*] \in \mathbb{R}^{\frac{N}{2} \times p}$ and $U_t[I_i^*] \in \mathbb{R}^{\frac{N}{2}}$ by choosing rows according to the index set $I_i^*$. For each of the resulting $B$ subproblems, apply a sparse regression technique and systematically record the recovered supports $\hat{S}^\lambda[I_i^*]$, $i = 1, \ldots, B$, as a function of $\lambda$ over a *regularization path* $\Lambda = [\lambda_{\max}, \lambda_{\min}]$. The values of $\lambda_{\max}$ and $\lambda_{\min}$ are data-dependent and are easily computable for generalized linear models with convex penalties [41]. In our case, the parameter $\lambda_{\max}$ for the non-convex problem in equation (2.8) can be determined from optimality conditions (Theorem 12 in [55] and Theorem 1 in [36]). The lower bound $\lambda_{\min}$ is set to $\lambda_{\min} = \epsilon \lambda_{\max}$ with default $\epsilon = 0.1$. The $\lambda$-dependent *importance measure* for each coefficient $\xi_k$ is then computed as

$$\hat{\Pi}_k^\lambda = \mathbb{P}(k \in \hat{S}^\lambda) \approx \frac{1}{B} \sum_{i=1}^{B} \mathbb{1}(k \in \hat{S}^\lambda[I_i^*]), \tag{2.10}$$

where $I_1^*, \ldots, I_B^*$ are the independent random sub-samples. The importance measure $\hat{\Pi}_k^\lambda$ of each model coefficient can be plotted across the regularization path, resulting in a component stability profile (see figure 1f for an illustration). This visualization provides an intuitive overview of the importance of the different model components. Different from the original stability selection work [33], we define the stable components of the model as

$$\hat{S}_{\text{stable}} = \{k : \hat{\Pi}_k^{\lambda_{\min}} \geq \pi_{\text{th}}\}. \tag{2.11}$$

Here, $\pi_{\text{th}}$ denotes the stability threshold parameter, which can be set to $\pi_{\text{th}} \in [0.7, 0.9]$ [33]. We always use the default setting $\pi_{\text{th}} = 0.8$. During exploratory data analysis, the threshold $\pi_{\text{th}}$ can also be set through visual inspection of the stability plots, allowing principled exploration of alternative PDE models. The importance measures $\hat{\Pi}_k^\lambda$ also provide an interpretable criterion for a model component's stability against random sub-sampling of the data and changes to the dictionary design, guiding the user to build the right model with high probability. As we show in the numerical experiments, stability selection thus ensures robustness against varying dictionary size, different types of data sampling, noise in the data and variability of the sub-optimal solutions when non-convex penalties are used. All of these properties are critical for consistent and reproducible model learning in real-world applications. Under certain conditions, stability selection also provides an upper bound on the expected number of false positives [33]. Such guarantees are not generally assured by any sparsity-promoting regression method in isolation [34]. For instance, stability selection combined with randomized LASSO (equation (2.7) with $\alpha < 0.5$) is consistent for variable selection even when the irrepresentable condition is violated [33].

## 3. Numerical experiments on simulated data

We present numerical experiments in order to benchmark the performance and robustness of PDE-STRIDE combined with different sparsity-promoting regression methods to infer PDEs from spatio-temporal data. To provide comparisons and benchmarks, we first use simulated data obtained by numerically solving known ground-truth PDEs, before applying our method to a real-world dataset from biology. The benchmark experiments on simulation data are presented in four subsections that demonstrate different aspects of the inference framework: §a demonstrates the use of different sparsity-promoting regression methods in our framework in a simple one-dimensional Burgers problem. Section b then compares their performance in order to choose the

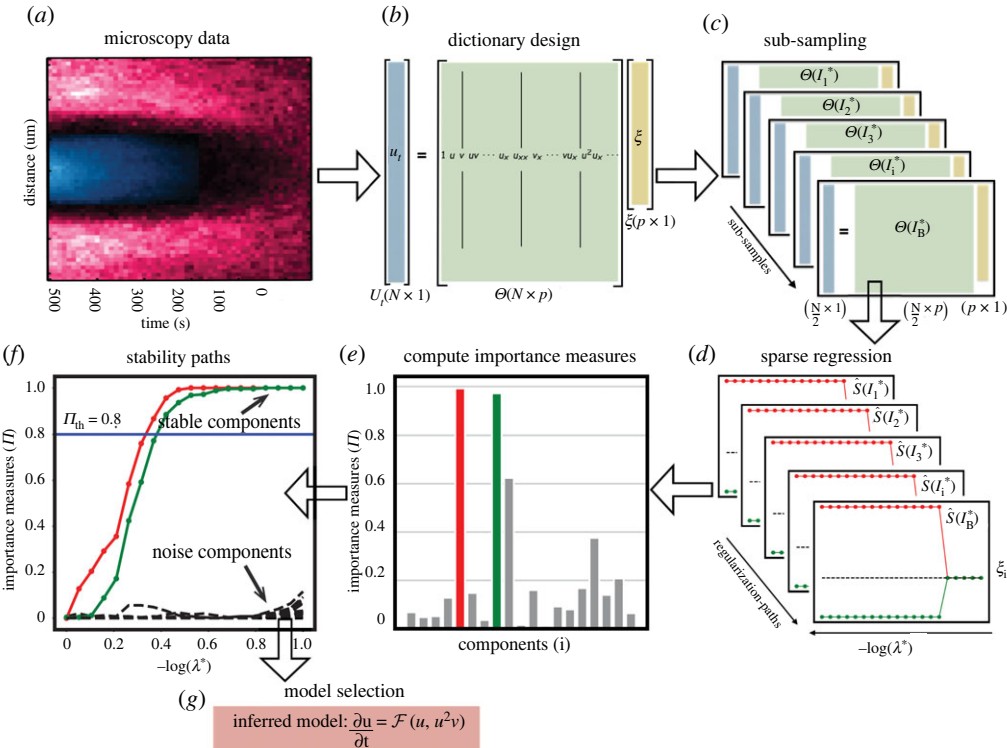

**Figure 1.** Enabling data-driven mathematical model discovery through stability selection. We outline the necessary steps in our method for learning PDE models from spatio-temporal data. (*a*) Extract spatio-temporal profiles from microscopy videos of the chosen state variables. Data courtesy of Grill Laboratory, MPI-CBG/TU Dresden [66]. (*b*) Compile the design matrix $\Theta$ and the measurement vector $U_t$ from the data. (*c*) Construct multiple linear systems of reduced size through random sub-sampling of the rows of the design matrix $\Theta$ and $U_t$. (*d*) Solve and record the sparse/penalized regression solutions independently for each sub-sample along the $\lambda$-paths. (*e*) Compute the importance measure $\Pi$ for each component. The histogram shows the importance measure $\Pi$ for all components at a particular value of $\lambda$. (*f*) Construct the stability plot by aggregating the importance measures along the $\lambda$-path, leading to separation of the noise variables (dashed black) from the stable components (coloured). Identify the most stable components by thresholding $\Pi > 0.8$. (*g*) Build the PDE model from the identified components. (Online version in colour.)

best regression method, IHT-d. In §c, stability selection is combined with IHT-d to recover two-dimensional vorticity-transport and three-dimensional reaction–diffusion PDEs from limited, noisy simulation data. Section d reports achievability results to quantify the robustness of stability selection to variations in dictionary size, sample size and noise levels. In all cases, we follow the PDE-STRIDE procedure as explained in the box below.

---

*STability-based Robust IDEntification of PDEs (PDE-STRIDE).* Given the noise-corrupted data $\tilde{u}$ and a choice of regression method, e.g. (randomized) LASSO, IHT, HTP, IHT-d, STRidge:

(i) Apply any required denoising method on the noisy data and compute the spatial derivatives and nonlinearities to construct the design matrix $\Theta \in$

$\mathbb{R}^{N \times p}$ and the time-derivatives vector $U_t \in \mathbb{R}^{N \times 1}$ for suitable sample size and dictionary size, $N$ and $p$, respectively.

(ii) Build the sub-samples $\Theta[I_i^*] \in \mathbb{R}^{N/2 \times p}$ and $U_t[I_i^*]$, for $i = 1, 2, \ldots, B$, by uniformly randomly sub-sampling of rows from the design matrix $\Theta$ and the corresponding rows from $U_t$. For every sub-sample $I_i^*$, standardize the sub-design matrix $\Theta[I_i^*]$ such that $\sum_{j=1}^{N/2} \theta_{jk} = 0$ and $(1/N) \sum_{j=1}^{N/2} \theta_{jk}^2 = 1$, for $k = 1, 2, \ldots, p$. Here, $\theta_{jk}$ is the element in row $j$ and column $k$ of the matrix $\Theta[I_i^*]$. The corresponding measurement vector $U_t[I_i^*]$ is centred to zero mean.

(iii) Apply the sparsity-promoting regression method independently to each sub-sample $\Theta[I_i^*], U_t[I_i^*]$ to construct the $\lambda$-paths for $M$ values of $\lambda$ as discussed in §2c.

(iv) Compute the importance measures $\hat{\Pi}_k^\lambda$ of all dictionary components $\xi_k$ along the discretized $\lambda$-paths, as discussed in §c. Select the stable support set $\hat{S}_{\text{stable}}$ by applying the threshold $\pi_{\text{th}} = 0.8$ to all $\hat{\Pi}_k$. Solve a linear least-squares problem restricted to the support $\hat{S}_{\text{stable}}$ to identify the coefficients of the learned model.

### (i) Adding noise to the simulation data

Let $u \in \mathbb{R}^N$ be the vector of clean simulation data sampled in both space and time. This vector is corrupted with additive Gaussian noise to

$$\tilde{u} = u + \varepsilon,$$

such that $\varepsilon = \sigma \mathcal{N}(0, \text{Var}(u))$ is the additive Gaussian noise with an empirical variance of the entries in the vector $u$, and $\sigma$ is the level of Gaussian noise added.

### (ii) Fixing the parameters for stability selection

We propose that PDE-STRIDE combined with IHT-d provides a PDE learning method that only rarely requires parameter tuning. To demonstrate this, all stability selection parameters described in §c are fixed to their standard values throughout our numerical experiments. The choice of these standard parameter values is well-discussed in the literature [33,41,64]. We thus fix: the repetition number $B = 250$, regularization path parameter $\epsilon = 0.1$, $\lambda$-path size $M = 20$ and the importance threshold $\pi_{\text{th}} = 0.8$. Using these standard values, the method works robustly across all tests presented here with no case-specific parameter tuning required. In both stability and regularization plots, we show normalized values of the regularization parameter $\lambda^* = \lambda/\lambda_{\max}$ on a decimal logarithmic scale. Although the stable component set $\hat{S}_{\text{stable}}$ is always evaluated at $\lambda_{\min} = 0.1\lambda_{\max}$, as in equation (2.11), the stability plots sometimes have their axes scaled to large or smaller ranges of $\lambda^*$ for better visualization.

## (a) One-dimensional Burgers equation with different sparsity promoters

We show that stability selection can be combined with any sparsity-promoting penalized regression framework to learn PDE components from noisy and limited spatio-temporal data. We use simulated data of the one-dimensional Burgers equation

$$\frac{\partial u}{\partial t} + u \frac{\partial u}{\partial x} = D \frac{\partial^2 u}{\partial x^2}, \tag{3.1}$$

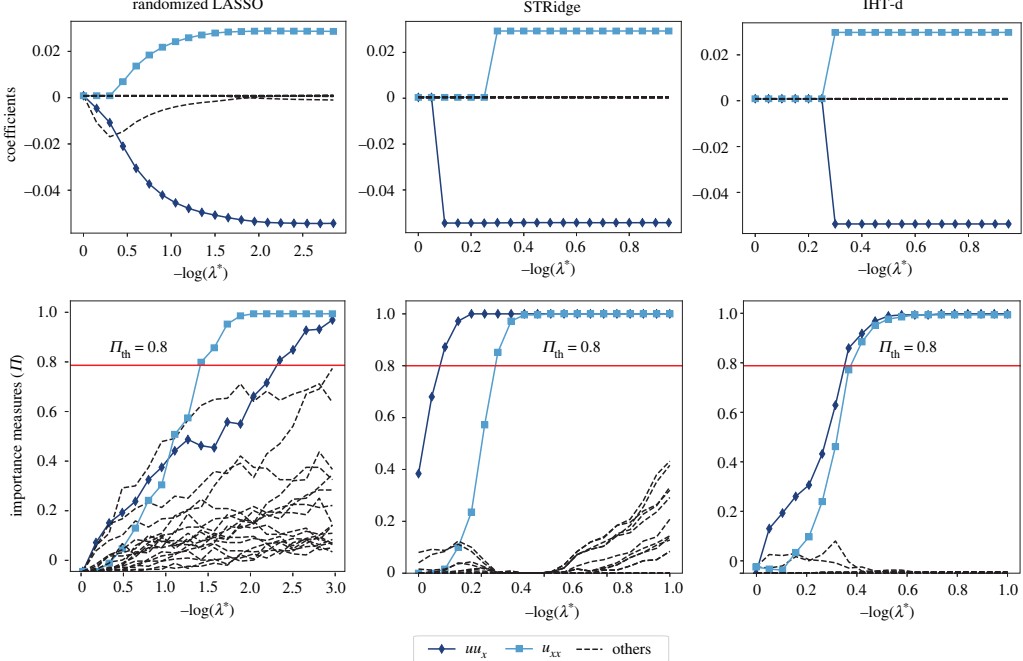

**Figure 2.** Model selection with PDE-STRIDE for the one-dimensional Burgers equation. The top row shows regularization paths (see §c) for three sparsity-promoting regression techniques: randomized LASSO, STRidge and IHT-d all for the same design ($N = 200, p = 19$). The bottom row shows the corresponding stability plots. The inset legend at the bottom shows the colour and line symbol correspondence with the dictionary. The ridge parameter $\lambda_R$ for STRidge is fixed to $\lambda_R = 10^{-5}$ [24]. The value of $\alpha$ for the randomized LASSO is set to 0.2. In all three cases, the standard threshold $\pi_{\text{th}} = 0.8$ (red solid horizontal line) correctly identifies the true components. The $\epsilon$ is set to 0.001 for randomized LASSO in order to demonstrate stability selection. (Online version in colour.)

with identical boundary and initial conditions as used in [24] to provide fair comparison between methods: periodic boundaries in space and the following Gaussian initial condition:

$$u(x, 0) = e^{-(x+2)^2}, \quad x \in [-8, 8].$$

The simulation domain $[-8, 8]$ is divided uniformly into 256 Cartesian grid points in space and 1000 time points. The numerical solution is visualized in space-time in figure 4. The numerical solution was obtained using a parabolic method based on finite differences and time stepping using explicit Euler with step size $dt = 0.01$ and the diffusion coefficient $D = 0.1$.

We test the combinations of stability selection with the three sparsity-promoting regression techniques described in §2b: randomized LASSO, STRidge and IHT-d. The top row of figure 2 shows the regularization paths and the bottom row the corresponding stability plots for each component in the dictionary. The coloured solid lines correspond to the advection and diffusion terms of equation (3.1) as given in the inset legend.

Thresholding the importance measure at $\Pi > \pi_{\text{th}} = 0.8$, all sparsity-promoting regression methods are able to identify the correct components of the model and separate them from the noise variables (dashed black lines).

## (b) Comparison between sparsity-promoting techniques

Although stability selection can be used in conjunction with any $\ell_1$ or $\ell_0$ sparsity-promoting regression method, the question arises whether a particular choice of regression algorithm is particularly well suited for PDE learning. We therefore perform a systematic comparison between

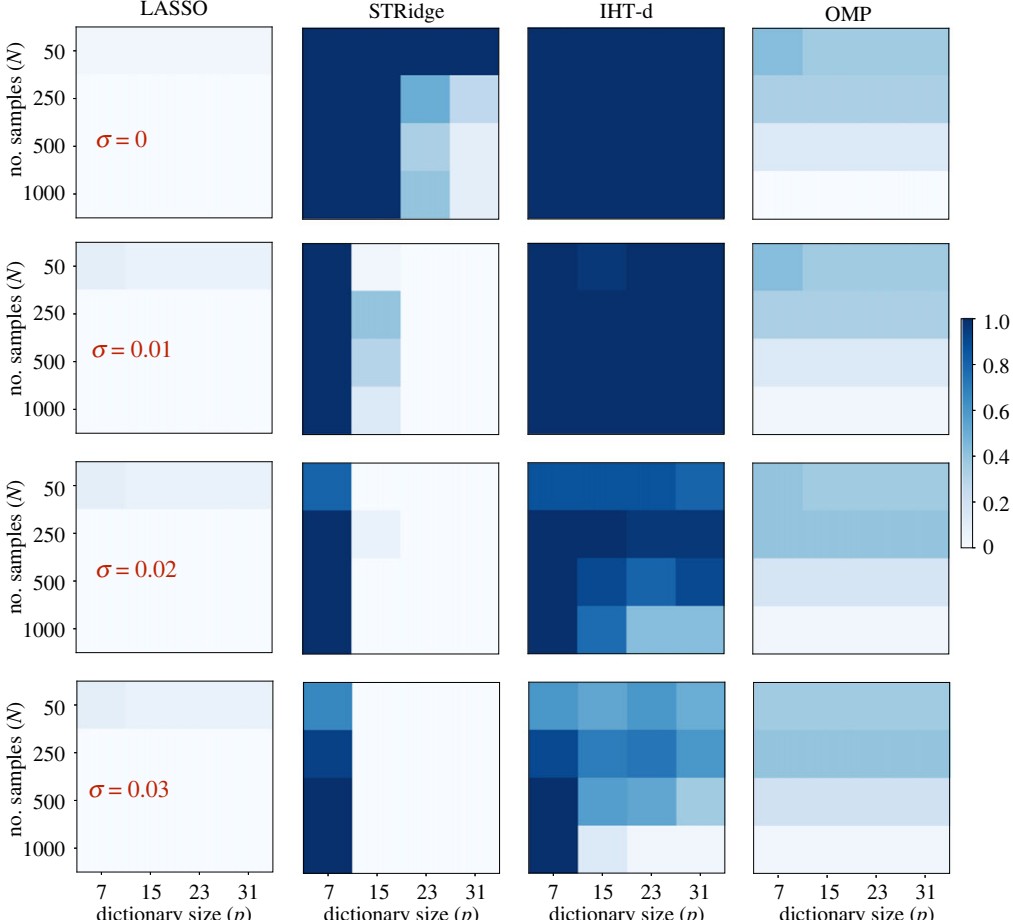

**Figure 3.** Comparison between different sparse regression methods for the one-dimensional Burgers equation. Each coloured square corresponds to a design ($N, p, \sigma$) with certain sample size $N$, dictionary size $p$ and noise level $\sigma$. Colour indicates the success frequency over 30 independent repetitions with uniformly random data samples, as given in the colour bar to the right. 'Success' is defined as the existence of $a$ $\lambda$ value for which the correct PDE is recovered from the data. The columns compare four popular sparsity-promoting regression methods: randomized LASSO, STRidge, IHT-d and OMP (left to right), as labelled at the top. (Online version in colour.)

randomized LASSO, STRidge, orthogonal matching pursuit (OMP) [67,68] as implemented in *scikit-learn*, and IHT-d for recovering the one-dimensional Burgers equation under perturbations to the sample size $N$, the dictionary size $p$ and the noise level $\sigma$. An experiment for a particular triple ($N, p, \sigma$) is considered a success if there exists a $\lambda \in \Lambda$ (see §c) for which the true PDE components are recovered. In figure 3, the success frequencies over 30 independent repetitions with uniformly random data sub-samples are shown for the four regression methods.

A first observation from figure 3 is that $\ell_0$ solutions (here with STRidge, IHT-d and OMP) outperform relaxed $\ell_1$ solutions (here with randomized LASSO). We also observe that IHT-d performs better than the other three methods for large dictionary sizes $p$, high noise $\sigma$ and small sample sizes $N$. Large dictionaries with higher-order derivatives computed from discrete data cause grouping (correlations between variables), for which randomized LASSO tends to select one variable from each group, ignoring the others [69]. Thus, randomized LASSO fails to identify the true support consistently. STRidge shows good recovery for large dictionary sizes $p$ with clean

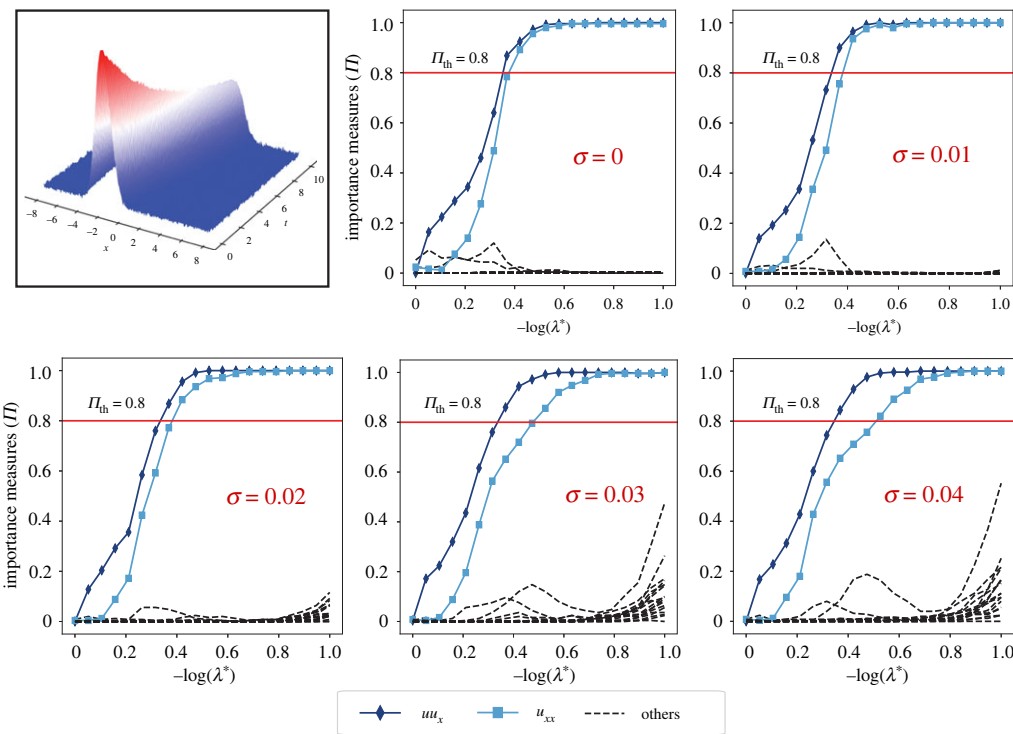

**Figure 4.** Model selection with PDE-STRIDE+IHT-d for the one-dimensional Burgers equation. The top left image shows the numerical solution of the one-dimensional Burgers equations on 256 × 1000 space and time points, respectively. The stability plots for the design $N = 250$, $p = 19$ show clear separation of the true PDE components (in solid colour with line symbols as identified in the inset legend) from the noise components (dashed black). The inference power of the method is tested for additive Gaussian noise levels $\sigma$ up to 4%. In all cases, perfect recovery is possible without parameter tuning. (Online version in colour.)

data, but it breaks down in the presence of noise in the data. OMP also uses $\ell_0$ regularization and therefore outperforms randomized LASSO, but fails to compete with STRidge and IHT-d. Of all four methods, IHT-d shows the best robustness to both noise and changes in the design. We note a decrease in inference power with increasing sample size $N$, especially for large $p$ and high noise levels. This again can be attributed to correlations and groupings in the dictionary, which become more significant with increasing sample size $N$.

Based on these results, we use IHT-d in the following sections as the sparsity-promoting regression method in PDE-STRIDE.

## (c) Stability-based model inference

We present benchmark results of PDE-STRIDE for PDE learning with IHT-d as the sparse regression method. This combination of methods is used to recover PDEs from limited noisy data obtained by numerical solution of the one-dimensional Burgers, two-dimensional vorticity-transport and three-dimensional Gray–Scott equations. Once the stable support $\hat{S}_{\text{stable}}$ of the PDE model has been learned by PDE-STRIDE with IHT-d, the actual coefficient values of the non-zero components are determined by solving a linear least-squares problem restricted to the recovered support $\hat{S}_{\text{stable}}$. More sophisticated methods could instead be used for PDE parameter estimation with known model structure [21,22]. As the sample size $N$ exceeds the cardinality of the recovered support ($N \gg |\hat{S}_{\text{stable}}|$), however, we find that simple least-squares fits provide good estimates for the PDE coefficients.

**Table 1.** Coefficient values of the recovered one-dimensional Burgers equation for different noise levels. The stable components of the PDE inferred from the plots in figure 4 are $\hat{S}_{\text{stable}} = \{u_{xx}, uu_x\}$. The ground-truth coefficient values are given in parentheses in the column headings.

| | $uu_x$ (−1.0) | $u_{xx}$ (0.1) |
|---|---|---|
| clean | −1.0008 | 0.1000 |
| 1% | −0.9971 | 0.1016 |
| 2% | −0.9932 | 0.0997 |
| 3% | −0.9842 | 0.0976 |
| 4% | −0.9728 | 0.0984 |
| 5% | −0.9619 | 0.0967 |

### (i) One-dimensional Burgers equation

We again consider the one-dimensional Burgers equation from equation (3.1), using the same simulated data as in §a, to quantify the performance and robustness against noise of the PDE-STRIDE+IHT-d method. The results are shown in figure 4 for a design with $N = 250$ and $p = 19$. Even on this small dataset, with a sample size comparable with dictionary size, our method recovers the correct model $\{u_{xx}, uu_x\}$ with up to 4% noise on the data, although the least-squares fits of the coefficient values gradually deviate from their exact values (table 1).

For comparison, the corresponding stability plots for PDE-STRIDE + STRidge are shown in the electronic supplementary material, figure S3. STRidge creates many false positives even at mild noise levels (less than 2%).

### (ii) Two-dimensional vorticity transport equation

Next, we consider a two-dimensional domain and the vorticity transport equation given in equation (3.2). The vorticity transport equation can be obtained by taking curl of the Navier–Stokes equations and imposing the incompressibility constraint $\nabla \cdot u = 0$ for the flow velocity field $u$. This results in a governing equation for the vorticity $\omega = \nabla \times u$, which is a scalar in two dimensions

$$\frac{\partial \omega}{\partial t} + u \frac{\partial \omega}{\partial x} + v \frac{\partial \omega}{\partial y} = \mu \left( \frac{\partial^2 \omega}{\partial x^2} + \frac{\partial^2 \omega}{\partial y^2} \right). \tag{3.2}$$

This equation has numerous applications in oceanography and climate modelling [70]. The velocity vector $u = (u, v)$ has two components, and $\mu$ is the fluid's viscosity. For the numerical solution of the transport equation (3.2) in the unit square, we impose a no-slip boundary condition at the left $(x = 0, y \in [0, 1])$, right $(x = 1, y \in [0, 1])$ and bottom sides $(y = 0, x \in [0, 1])$ and a shear flow boundary condition with boundary velocity components $u_{\text{boundary}} = 2.0$, $v_{\text{boundary}} = 0$ on the top side $(y = 1, x \in [0, 1])$. This is the classic 'lid-driven cavity' problem from fluid mechanics. The simulation code was written using OpenFPM [71] with explicit time stepping on a $128 \times 128$ grid in space. A Poisson equation was solved at every time step in order to correct the velocities to ensure divergence-freeness. The viscosity of the fluid simulated was set to $\mu = 0.025$. In figure 5, we show a single time snapshot of the two velocity components $(u, v)$ and the vorticity $\omega$ in the numerical simulation, along with the locations of the 500 sample points used for PDE inference from these data.

Figure 6 shows the results of applying PDE-STRIDE + IHT-d to recover the two-dimensional vorticity transport equation from these data samples. The results demonstrate consistent recovery of the true support of the PDE for different noise levels $\sigma$. The stable components $\hat{S}_{\text{stable}} = \{\omega_{xx}, \omega_{yy}, u\omega_x, v\omega_y\}$ correspond to the true terms of equation (3.2). In table 2, we show the re-fitted coefficients for the recovered PDE components. Both the accuracy of the parameter fits and the separation between the true (coloured solid lines) and the noisy (black dashed lines) components

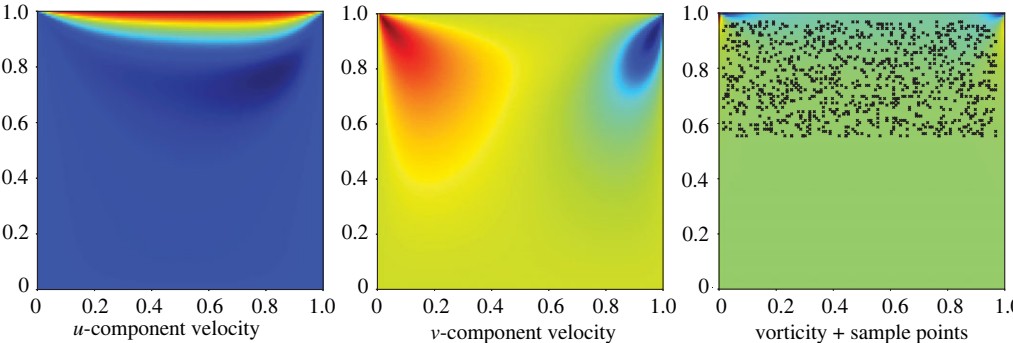

**Figure 5.** Numerical solution of the two-dimensional vorticity transport equation. We show the two components of the flow velocity $(u, v)$ and the vorticity $\omega$ in the two-dimensional domain $[0, 1]^2$. The black dots in the right panel show the $N = 500$ sample points used for PDE inference. They are uniformly randomly distributed in the rectangular box $[0.0, 1.0] \times [0.6, 1.0]$. (Online version in colour.)

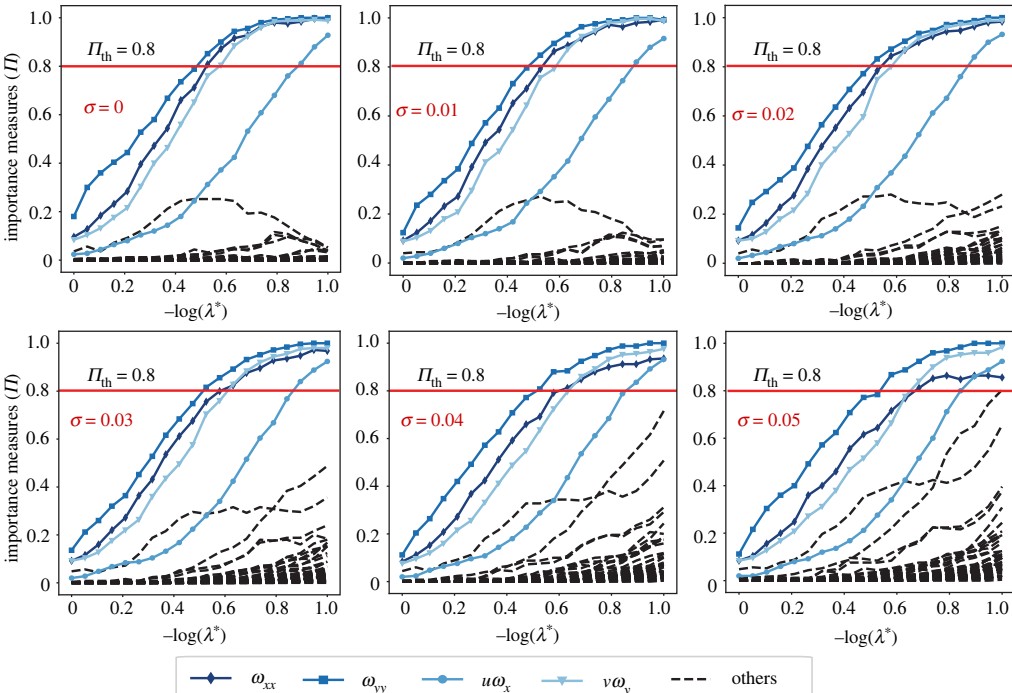

**Figure 6.** Model selection with PDE-STRIDE + IHT-d for the two-dimensional vorticity transport equation. The stability plots for the design $N = 500, p = 48$ show the separation of the true PDE components (in solid colour with line symbols as identified in the inset legend) from the noise components (dashed black). The inference power of the method is tested for additive Gaussian noise levels $\sigma$ up to 5%. In all cases, perfect recovery is possible without parameter tuning. (Online version in colour.)

deteriorate with increasing noise levels. In the electronic supplementary material, figure S4, we also report the plots when using STRidge in conjunction with stability selection for the same design and stability selection parameters. STRidge fails to recover the true support even at small noise levels.

**Table 2.** Coefficient values of the recovered two-dimensional vorticity transport equation for different noise levels. The stable components of the PDE inferred from the plots in figure 6 are $\hat{S}_{stable} = \{\omega_{xx}, \omega_{yy}, u\omega_x, v\omega_y\}$. The ground-truth coefficient values are given in parentheses in the column headings.

| | $\omega_{xx}$ (0.025) | $\omega_{yy}$ (0.025) | $u\omega_x$ (−1.0) | $v\omega_y$ (−1.0) |
|---|---|---|---|---|
| clean | 0.02504 | 0.02502 | −0.9994 | −1.0025 |
| 1% | 0.02501 | 0.02504 | −0.9997 | −1.0006 |
| 2% | 0.02492 | 0.0250 | −1.0003 | −0.9944 |
| 3% | 0.0247 | 0.0250 | −1.004 | −0.9841 |
| 4% | 0.0245 | 0.0251 | −1.0091 | −0.9748 |
| 5% | 0.0242 | 0.0251 | −1.0083 | −0.9586 |

### (iii) Three-dimensional Gray–Scott equation

Finally, we consider a problem in three dimensions, namely, the three-dimensional Gray–Scott reaction–diffusion model

$$\frac{\partial u}{\partial t} = D_u \left( \frac{\partial^2 u}{\partial x^2} + \frac{\partial^2 u}{\partial y^2} + \frac{\partial^2 u}{\partial z^2} \right) - uv^2 + f(1-u) \tag{3.3a}$$

and

$$\frac{\partial v}{\partial t} = D_v \left( \frac{\partial^2 v}{\partial x^2} + \frac{\partial^2 v}{\partial y^2} + \frac{\partial^2 v}{\partial z^2} \right) + uv^2 - (f+k)v. \tag{3.3b}$$

Reaction and diffusion of two chemical species with scalar concentrations $u$ and $v$ can produce a variety of patterns, reminiscent of those often observed in nature [72]. This is, for example, used to describe skin patterning and pigmentation [73]. This example also has coupled variables and is similar in structure to the real-world example discussed in §4.

We simulate equation (3.3) using second-order central finite differences implemented in OpenFPM [71]. A snapshot of the simulated concentration field $u$ in the three-dimensional cube $[0, 2.5]^3$ is shown in figure 7. The simulation used $128 \times 128 \times 128$ discretization points in space on a regular Cartesian mesh and explicit Euler time stepping with step size $dt = 0.0005$ until a final simulated time of 5 seconds. The ground-truth model parameters used are: $k = 0.053$, $f = 0.014$, $D_u = 2.0 \times 10^{-5}$ and $D_v = 10^{-5}$.

We test recovery of the ground-truth PDE from data sampled only from the small cube $[1.0, 1.5]^3$ in the centre of the domain. Figure 7 shows the PDE-STRIDE+IHT-d results for the species $u$. All PDE components of the true equation (3.3a) are correctly identified for noise levels up to 6% with as few as $N = 400$ samples for dictionary size $p = 69$. The plots for the $v$ species are shown in the electronic supplementary material, figure S5. Although perfect recovery was not possible owing to the small diffusivity ($D_v = 10^{-5}$) of the $v$ species, consistent and stable recovery of the reaction terms is seen. The re-fitted coefficients in the recovered PDEs for the $u$ and $v$ species are reported in table 3 and the electronic supplementary material, table S1, respectively.

For comparison, the results when using STRidge in conjunction with stability selection are shown in the electronic supplementary material, figures S6 and S7. STRidge is able to recover the complete form of equation (3.3), albeit only in the noise-free case. It fails to recover any of the two components when noise is added to the data.

### (d) Achievability results

We discuss the consistency and robustness of PDE-STRIDE + IHT-d with respect to design parameters including sample size $N$, dictionary size $p$ and noise level $\sigma$. Achievability analysis provides a compact way of checking robustness and consistency of a model selection method for varying design parameters. It also provides approximate means to reveal the *sample complexity* of

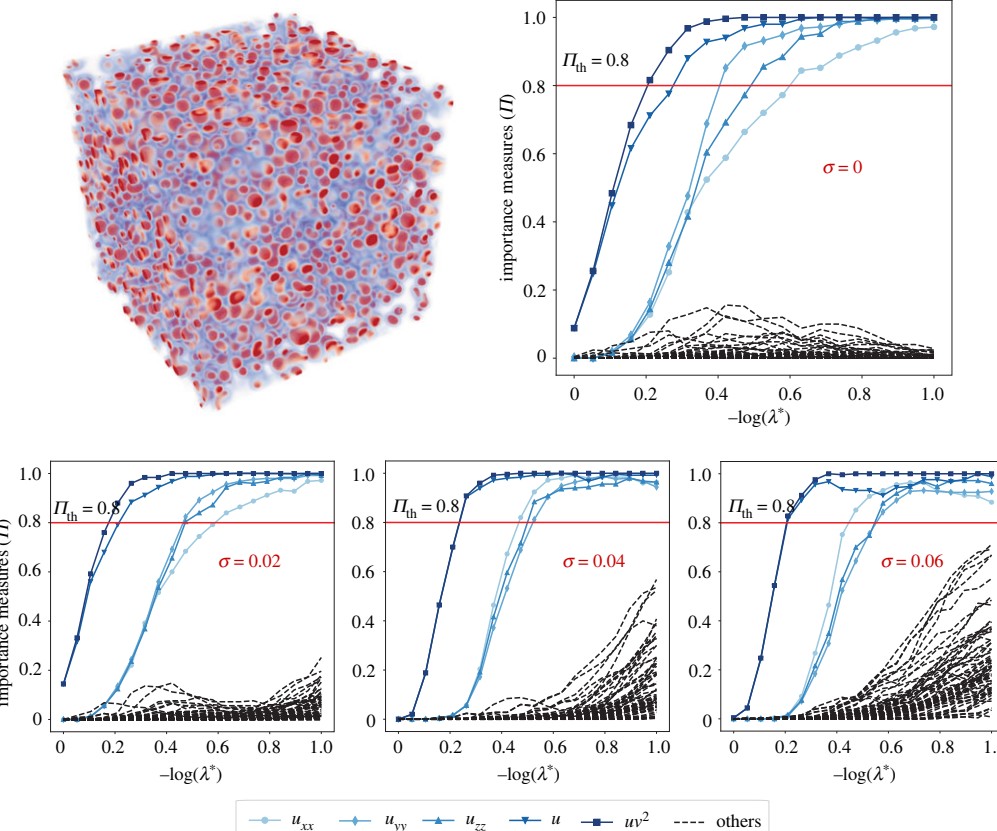

**Figure 7.** Model selection with PDE-STRIDE+IHT-d for the three-dimensional Gray–Scott $u$-component equation. The top left figure shows a visualization of the scalar concentration field $u$ in the three-dimensional simulation domain with red corresponding to high concentration and blue to low concentration. The stability plots for the design $N = 400$, $p = 69$ show good separation of the true PDE components (in solid colour with line symbols as identified in the inset legend) from the noise components (dashed black). The inference power of the method is tested for additive Gaussian noise levels $\sigma$ up to 6%. In all the cases, perfect recovery was possible without parameter tuning. (Online version in colour.)

**Table 3.** Coefficients of the recovered $u$-component Gray–Scott reaction–diffusion equation for different noise levels. The stable components of the PDE inferred from the plots in figure 7 are $\hat{S}_{\text{stable}} = \{u_{xx}, u_{yy}, u_{zz}, u, uv^2\}$. The ground-truth coefficient values are given in parentheses in the column headings.

| | 1 (0.014) | $u_{xx}$ (2.0 × 10$^{-5}$) | $u_{yy}$ (2.0 × 10$^{-5}$) | $u_{zz}$ (2.0 × 10$^{-5}$) | $u$ (−0.014) | $uv^2$ (−1.0) |
|---|---|---|---|---|---|---|
| clean | 0.0140 | 2.0 × 10$^{-5}$ | 2.0 × 10$^{-5}$ | 2.0 × 10$^{-5}$ | −0.0140 | −1.0000 |
| 2% | 0.0142 | 1.9664 × 10$^{-5}$ | 1.9565 × 10$^{-5}$ | 1.9869 × 10$^{-5}$ | −0.0143 | −0.9915 |
| 4% | 0.0144 | 1.9541 × 10$^{-5}$ | 1.8971 × 10$^{-5}$ | 1.8780 × 10$^{-5}$ | −0.0146 | −0.9795 |
| 6% | 0.0150 | 2.0494 × 10$^{-5}$ | 1.8888 × 10$^{-5}$ | 1.8284 × 10$^{-5}$ | −0.0153 | −0.9843 |

any $\ell_0$ and $\ell_1$ sparsity-promoting technique, i.e. the number of data points $N$ required to recover the model with full probability. Specifically, given a sparsity-promoting regularizer, dictionary size $p$, sparsity $k$ and noise level $\sigma$, we are interested in how the sample size $N$ scales with $p, k, \sigma$ for the recovery probability converging to one. The study in [74] reported sharp phase transitions

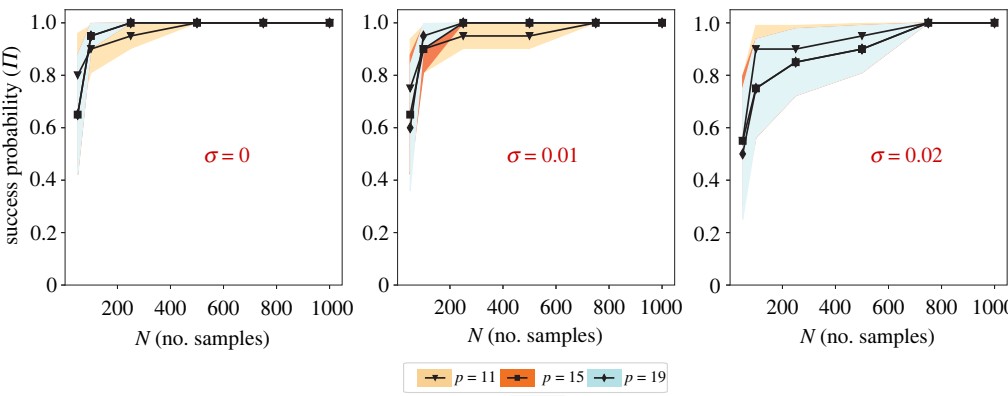

**Figure 8.** Achievability plots for model selection with PDE-STRIDE+IHT-d for the one-dimensional Burgers equation. Each symbol is the mean over 20 repetitions of the inference for some design $(N, p, \sigma)$ under random data sub-sampling. Different symbols and colours correspond to different dictionary sizes $p$ as given in the inset legend. The coloured bands show the variance of the Bernoulli trials. Noise levels $\sigma$ are given by the red percentage in each panel. (Online version in colour.)

from failure to success for Gaussian random designs with increasing sample size $N$ for LASSO-based sparsity solutions. The same study also provided sufficient lower bounds for sample size $N$ as a function of $p, k$ for full recovery probability. We ask the question whether sparse model selection with PDE-STRIDE + IHT-d exhibits similar sharp phase-transition behaviour. Given the dictionary components in PDE learning are compiled from derivatives and nonlinearities computed from noisy data, it is also interesting to observe whether full recovery is at all achieved and maintained with increasing sample size $N$. In the particular context of PDE learning, increasing dictionary size by including higher-order nonlinearities and higher-order derivatives tends to introduce strongly correlated components, which can negatively impact the inference power.

In figures 8 and 9, the achievability plots for the one-dimensional Burgers system and the $u$-component of the three-dimensional Gray–Scott system are shown, respectively. Each symbol in the figures shows the mean over 20 repetitions of an experiment with some design $(N, p, \sigma)$ under random data sub-sampling. The Bernoulli variances are shown as coloured bands. An experiment with a design $(N, p, \sigma)$ is considered a success if and only if there exists a $\lambda \in \Lambda$ for which the true PDE support is recovered by PDE-STRIDE+IHT-d with default importance threshold $\pi_{th} = 0.8$.

In all cases, PDE-STRIDE + IHT-d is strongly consistent and highly robust. In addition, we also observe a sharp phase transition from failure to success with recovery probabilities quickly approaching one for increasing sample size beyond a certain threshold. This suggests that PDE-STRIDE not only enhances the inference power of IHT-d but also ensures consistency. The sharp phase transition also suggests the existence of a strict lower bound on the sample complexity, below which full recovery is not achievable [74]. From the achievability plots, we estimate the sample complexity of the learned dynamical systems: for the one-dimensional Burgers equation, 90% success probability is achieved with as few as $\approx 70$ data points in the noise-free and $\approx 200$ data points in the noisy cases across different designs ($p$). For the three-dimensional Gray–Scott system, 90% success probability is achieved with as few as $\approx 200$ data points in the noise-free and $\approx 400$ in the noisy cases across different designs ($p$). This demonstrates that PDE-STRIDE + IHT-d is able to consistently and robustly learn PDE models from limited noisy data.

## 4. Data-driven inference on experimental data to explain *C. elegans* zygote polarization

We showcase the applicability of the PDE-STRIDE with IHT-d to real experimental data. We use microscopy images to infer a model that explains early *C. elegans* embryo polarity

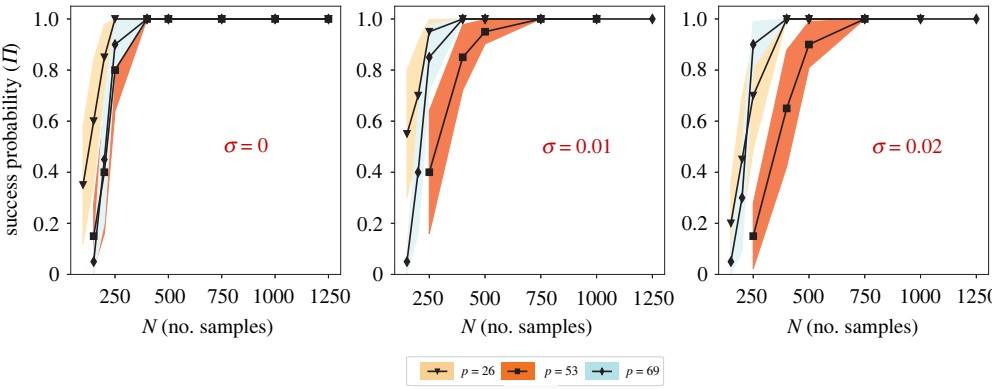

**Figure 9.** Achievability plots for model selection with PDE-STRIDE+IHT-d for the *u*-component of the three-dimensional Gray–Scott equation. Each symbol is the mean over 20 repetitions of the inference for some design ($N$, $p$, $\sigma$) under random data sub-sampling. Different symbols and colours correspond to different dictionary sizes $p$ as given in the inset legend. The coloured bands show the variance of the Bernoulli trials. Noise levels $\sigma$ are given by the red percentage in each panel. (Online version in colour.)

establishment, and we use it to confirm a previous hypothesis about the underlying protein interaction network. Earlier studies of this process proposed a mechano-chemical mechanism for PAR protein polarization on the cell membrane [15,16,66]. They systematically showed that active cortical flows in the zygote provide sufficient bias to trigger symmetry breaking [16]. The experiments conducted in [16] measured the concentration of the anterior PAR complex (aPAR), the concentration of the posterior PAR complex (pPAR) and the cortical flow velocity field as a function of time (figure 10*a–d*). The concentration and velocity fields were acquired on a grid with a resolution of $60 \times 55$ in space and time, respectively. Space was modelled one-dimensional along a circumferential line around the embryo in the mid-plane cross section as shown in figure 10*a*.

We challenge PDE-STRIDE + IHT-d to learn a differential equation model for the regulatory network of the interacting PAR proteins in a purely data-driven fashion from the one single microscopy video available as electronic supplementary material for [16]. Given the high level of noise in the video, we limit our analysis to the first SVD mode of the data as shown in figure 10*c*. We only consider the time after the initial advection trigger, when early domains of PAR proteins have already formed. PDE-STRIDE is then directed to learn an interpretable model that evolves the nascent protein domains to the fully developed polarity patterns shown in figure 10*a*.

The only mechanism included in this model is the chemical interaction between aPAR and pPAR of the general form

$$v_a^- A + v_p^- P \xrightarrow{k} v_a^+ A + v_p^+ P, \tag{4.1}$$

with unknown reactant and product stoichiometries $v_{a/p}^-$ and $v_{a/p}^+$ for aPAR and pPAR, respectively. The scalar fields $A$ and $P$ are the concentrations of aPAR and pPAR, and $k$ is the unknown reaction rate.

In designing the dictionary $\Theta$, the maximum allowed stoichiometry for reactants and products is restricted to 2, i.e. $v_{a/p}^-, v_{a/p}^+ \in \{0, 1, 2\}$. The PDE-STRIDE + IHT-d results for the learned reaction network from data are shown in figure 10*e,f*. The stable components of the model for pPAR are $\hat{S}_{\text{stable}}^P = \{P, P^2 A\}$, and for aPAR they are $\hat{S}_{\text{stable}}^A = \{A, PA^2\}$ for $N \approx 500$, $p = 20$. The achievability plot in figure 10*g* confirms the consistency and robustness of the learned model across different sample sizes $N$. The inference method achieves full recovery probability for $N > 800$.

The final chemical reaction model learned by PDE-STRIDE + IHT-d is

$$\frac{\mathrm{d}P}{\mathrm{d}t} = c_P + k_P P + k_{PA} A P^2 \tag{4.2a}$$

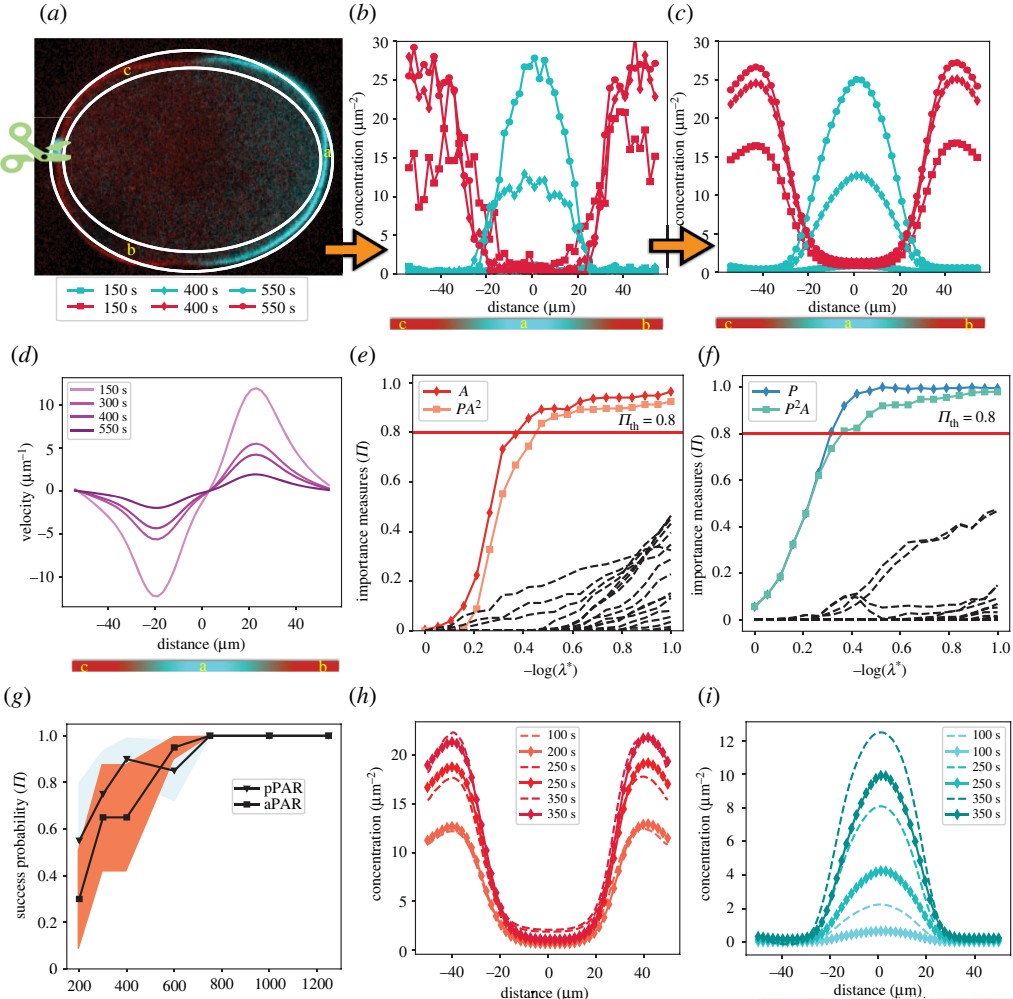

**Figure 10.** Data-driven inference of the regulatory network of PAR proteins from spatio-temporal data of a *C. elegans* zygote. (*a*) Final frame from a fluorescence microscopy video recorded by the Grill laboratory [16] at MPI-CBG showing a mid-plane cross section through the ellipsoidal 50 μm zygote. Over time, the pPAR protein complex (blue fluorescence signal) localizes to the posterior pole of the embryo (region 'a'), whereas aPAR (red fluorescence signal) localizes anterior (regions 'b' and 'c'). (*b*) Protein concentrations extracted from the fluorescence intensity along the one-dimensional midline between the two white ellipses in (*a*) for aPAR (red) and pPAR (blue) for different time points (symbols, see inset legend under (*a*)). (*c*) De-noised concentration profiles obtained by extracting the principle mode of the singular value decomposition (SVD) of the noisy data in (*b*). (*d*) De-noised cortical flow velocity profiles (first SVD mode) in the tangential direction along the circumferential line around the zygote for different times (colour, inset legend). (*e,f*) Stability plots obtained by using PDE-STRIDE + IHT-d to separate the stable model components (coloured solid lines with symbols, inset legend) from the noise components (black dashed lines) for aPAR (*e*) and pPAR (*f*). (*g*) Achievability plot to test the robustness and consistency of the inferred model for both aPAR ($S_{stable}^{A} = \{A, A^2P\}$) and pPAR ($S_{stable}^{P} = \{P, P^2A\}$) with increasing sample size $N$. (*h,i*) Simulation results (dashed lines) of the learned models for aPAR (*h*) and pPAR (*i*) compared with the denoised experimental data (solid lines with symbols) at different times (inset legend). (Online version in colour.)

and

$$\frac{\mathrm{d}A}{\mathrm{d}t} = c_A + k_A A + k_{AP} PA^2, \tag{4.2b}$$

where $k_A$ and $k_P$ are the kinetic rate constants for membrane association and dissociation of aPAR and pPAR, respectively. The coefficients $k_{PA}$ and $k_{AP}$ represent the rates of membrane dissociation

**Table 4.** Coefficients values of the inferred model in equation (4.2).

| $c_P$ | $-1.98 \times 10^{-4}\,\mu\text{m}^{-2}\,\text{s}^{-1}$ | $c_A$ | $4.13 \times 10^{-3}\,\mu\text{m}^{-2}\,\text{s}^{-1}$ |
|---|---|---|---|
| $k_P$ | $1.07 \times 10^{-2}\,\text{1 s}^{-1}$ | $k_A$ | $2.16 \times 10^{-3}\,\text{1 s}^{-1}$ |
| $k_{PA}$ | $-2.79 \times 10^{-4}\,\mu\text{m}^4\,\text{s}^{-1}$ | $k_{AP}$ | $-1.47 \times 10^{-4}\,\mu\text{m}^4\,\text{s}^{-1}$ |

due to mutually antagonistic interactions between the protein complexes. The constants $c_P$ and $c_A$ can be interpreted as surface-to-volume conversion factors for the concentrations [16]. This purely data-driven model recapitulates the mutual inhibitory nature of the PAR proteins [16].

Moreover, our method allows us to estimate the values of the unknown reaction-rate parameters by least-squares re-fitting on the recovered stable support. The results are shown in table 4.

In the figure 10*h,i*, we overlay the numerical solution of equation (4.2) (dashed lines) with the denoised experimental data (solid lines with symbols) at different time points (see inset legend) for both aPAR and pPAR. The de-noised spatio-temporal measurement data from the early domains are taken as the initial conditions for the simulation. The results show that the simple data-driven chemical reaction model is able to qualitatively match the temporal evolution of the PAR protein domains in the *C. elegans* zygote. Although the spatial patterns match well (e.g. the PAR domain sizes) for the two proteins, there exist non-negligible discrepancies between the simulation and the experiments in the time scales of the pPAR domain evolution. This is likely because our ODE model only includes the chemical reactions of the protein interactions, but neither the diffusion nor the advective flow of the proteins. In the electronic supplementary material, figure S8 (left), we show for $N = 500$, $p = 20$ that the advection and diffusion components of the aPAR species carry enough importance to be included in the stable set $\hat{S}^A_{\text{stable}}$, but that this is not the case for the pPAR components. The preferential advective displacement of aPAR to the anterior side modelled by the advective term $(v_x A)$ is also in line with experimental observations [16]. However, such models with advection and diffusion components exhibit inconsistency for increasing sample size $N$, in contrast to our simple ODE model, as illustrated in figure 10*g*. We therefore believe that it may be necessary to use structured sparsity for enforcing conservation laws through grouping [75] in order to further develop this data-driven model to also include the mechanical aspects of the PAR system.

# 5. Conclusion and discussion

We have addressed two key issues that have so far limited the application of sparse regression methods for automated PDE inference from noisy and limited data: the need for manual parameter tuning and the high sensitivity to noise in the data. We have shown that stability selection combined with any sparsity-promoting regression technique provides an appropriate level of regularization for consistent and robust recovery of the correct PDE model. Our numerical benchmarks suggested that iterative hard thresholding with de-biasing (IHT-d) is ideal in combination with stability selection to form a robust framework for PDE learning with minimal parameter tuning. This combination of methods outperformed all other tested algorithmic approaches with respect to identification performance, amount of data required and robustness to noise. The resulting stability-based PDE-STRIDE method was tested for robust recovery of the one-dimensional Burgers equation, two-dimensional vorticity transport equation and three-dimensional Gray–Scott reaction–diffusion equations from simulation data corrupted with up to 6% additive Gaussian noise. The achievability studies demonstrated the consistency and robustness of the PDE-STRIDE method for full recovery probability of the model with increasing sample size $N$ and for varying dictionary size $p$ and noise levels $\sigma$. In addition, we confirmed the sharp phase transition in recovery performance and noted that achievability plots provide a natural estimate for the sample complexity of the underlying nonlinear dynamical

system. However, this empirical estimate of sample complexity depends on the choice of model selection scheme and on how the data are sampled.

We demonstrated the capabilities of PDE-STRIDE by learning a protein-interaction model of embryo polarization directly from fluorescence microscopy images of a *C. elegans* zygote. The model recovered the regulatory reaction network of the involved proteins, complete with its parameter values in a purely data-driven manner, with no knowledge used about the underlying physics or symmetries. The thus-learned, data-derived model was able to correctly predict the spatio-temporal dynamics of the embryonic polarity system from the early spatial domains to the fully developed patterns as observed in the polarized *C. elegans* zygote. The model we inferred from image data using our method confirms both the structure and the mechanisms of physics-derived cell polarity models [16]. Importantly, the mutually inhibitory interactions between the involved protein species, which have previously been discovered by extensive biochemical experimentation, were automatically and unambiguously extracted from the data [16].

Besides rendering sparse inference methods more robust to noise and to parameter settings, stability selection has the important conceptual benefit of also providing interpretable probabilistic importance measures for all model components. This enables modellers to construct their models with high fidelity, and to gain an intuition about correlations and sensitivities. Graphical inspection of stability plots provides additional freedom for intervention in semi-automated model discovery from data.

We expect that statistical learning methods have the potential to enable robust, consistent and reproducible discovery of predictive and interpretable models directly from observational data. Our PDE-STRIDE framework provides a first step towards this goal, but several open issues remain. First, numerically approximating time and space derivatives in the noisy data is a challenge for noise levels higher than a few percent. This currently limits the noise robustness of the overall method, regardless of how robust the subsequent statistical inference is. The impact of noise becomes even more severe when exploring models with higher-order derivatives or stronger nonlinearities. Future work should focus on formulations that are robust to the choice of different derivative-discretization methods, while providing the necessary freedom to impose structure on the coefficients. Here, signal/noise decomposition techniques based on deep neural networks with time-stepping constraints [76,77] could provide better denoising when the noise has finite correlation length in space and time.

Second, it would be desirable to have a principled way to constrain the learning process by physical priors, such as conservation laws and symmetries. Exploiting structural knowledge about the dynamical system is expected to greatly improve learning performance. Structured sparsity or grouping constraints from statistics may help express such prior knowledge in a sparse inference problem [75,78]. In the specific example of the PAR-polarity model, decades of experimentation and theory have revealed physical principles of mass conservation, detailed force balance in the cell cortex and antagonistic interactions between protein complexes. Using structured constraints to encode such physical principles leads to biologically plausible and physically consistent models, rather than models that simply fit the data [79].

In summary, we believe that data-driven model discovery has tremendous potential to provide novel insights into complex systems, in particular in biology. It provides an effective and complementary alternative to theory-driven approaches. We hope that the stability-based model selection method PDE-STRIDE presented here is going to contribute to the further development and adoption of these approaches in the sciences.

Data accessibility. The git repository for the codes and data can be found at https://git.mpi-cbg.de/mosaic/software/machine-learning/pde-stride. The data are provided in the electronic supplementary material [80].

Authors' contributions. S.M.: conceptualization, data curation, formal analysis, investigation, software, validation, visualization, writing—original draft; B.L.C.: data curation, formal analysis, resources, writing—review and editing; I.F.S.: conceptualization, funding acquisition, investigation, methodology, project administration, resources, supervision, writing—review and editing; C.L.M.: conceptualization, formal analysis, investigation, methodology, project administration, resources, supervision, validation, writing—review and editing.

All authors gave final approval for publication and agreed to be held accountable for the work performed therein.

Conflict of interest declaration. We declare we have no competing interests.

Funding. This work was in parts supported by the German Research Foundation (DFG, Deutsche Forschungsgemeinschaft) under funding code EXC-2068, Cluster of Excellence 'Physics of Life', and by the Center for Scalable Data Analytics and Artificial Intelligence (ScaDS.AI) Dresden/Leipzig funded by the Federal Ministry of Education and Research (BMBF, Bundesministerium für Bildung und Forschung).

Acknowledgements. We are grateful to the Grill laboratory at MPI-CBG, who provided us with high-quality spatial-temporal PAR concentration and flow field data. We also thank Nathan Kutz (University of Washington) and his group for making their code and data public.

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
