## [Peer Review File · Proceedings. Mathematical, Physical, and Engineering Sciences]

Review History

RSPA-2021-0916.R0 (Original submission)

Review form: Referee 1

Is the manuscript an original and important contribution to its field?

Good

Is the paper of sufficient general interest?

Acceptable

Is the overall quality of the paper suitable?

Good

Do you have any ethical concerns with this paper?

No

Recommendation?

Major revision is needed (please make suggestions in comments)

Comments to the Author(s)

The paper by Suryanarayana et al presents an approach for learning interpretable and robust PDE models from data. The topic is surely relevant; the method consists of an adaptation of regularised regression techniques (with L_0 regularisation) combined with a stability-based model selection criterion (as opposed to say a more traditional cross-validation error measure). The paper is well written and thoroughly evaluated; my impression is that, while it may not be radically innovative, it has certainly value and would be a good addition to the literature. My main comments are as follows:

- my understanding is that the IHT-d solver is a new heuristic, so it is not clear to me whether it inherits the guarantees cited by the authors for other algorithms. Which in turn may impact the "standard" choices for the parameters. Some clarification here would be useful.
- The PDE-STRIDE results are evaluated thoroughly on three simulated setups (perhaps two would suffice) plus one real example. All comparisons are however between different regularisers, I would also like to see in practice (maybe on only one example) practical advantages vis a vis an existing software tool.
- I would not go as far as calling PDE-STRIDE parameter free, I understand the qualifications but it seems a stretch.
- is the assumption of uncorrelated noise realistic? I would imagine that in real data there would be both spatial and temporal correlations in the noise. How would PDE stride work if the noise was e.g. a Gaussian process with a certain (perhaps small) spatial/ temporal correlation length? An experiment would be ideal but at least a discussion is needed.
- the real data application is somewhat underdeveloped. I would like to know whether the identified PDE makes biological sense. Additionally, shouldn't there be some conservation laws in place? How could these be enforced? Please also write out the final PDE system.

Review form: Referee 2

Is the manuscript an original and important contribution to its field?

Good

Is the paper of sufficient general interest?

Excellent

Is the overall quality of the paper suitable?

Excellent

Can the paper be shortened without overall detriment to the main message?

Yes

Do you think some of the material would be more appropriate as an electronic appendix?

No

Do you have any ethical concerns with this paper?

No

Recommendation?

Major revision is needed (please make suggestions in comments)

Comments to the Author(s)

In this paper the authors propose a methodology for inferring PDEs from noisy measurement data. They provide what I believe is a valuable contribution to an active research field by

combining stability selection and sparsity-promoting regression techniques. The paper contains some comparisons to other algorithms and experiments which suggest that this method can be usefully applied to real-world problems such as in the PAR example. All in all a solid research project that I can only recommend be published in this journal.

The reason for my recommending a major revision is that the RS requires all authors to make their supporting data, code and materials available. While the authors here provide a GitHub repository with some code, which includes the main algorithm, the only experiments they provide code for are on the Burgers equation (unless I am missing something). The code is generally understandable, although the second half of the iterative solver file could benefit from more functional encapsulation.

Furthermore the first page of Section 3b) appears empty on my computer. Judging from the first sentence on page 12 there is some content missing.

Minor comments:

- Fig. 1: The main font (Comic Sans?) is not very legible, and smaller labels are too thin. I would consider scaling up everything and removing the black border.
- L8. Remove comma after "models". As it stands it sounds like predictive models have to be derived from first principles and symmetry arguments, which clashes with the remaining paragraph.
- The last paragraph of 2b is a bit unclear on the approximate debiasing. The upper bound should ideally be specified in the main text as otherwise it is difficult to follow the reasoning without looking at the SI.
- In the PAR example the translation from model components (P, P2A etc.) to mechanistic components should be made more clear.
- In Figures S6 and S7 it seems strange that all parameters end up with a very high importance measure. Neither figure suggests that perfect recovery is the case, contrary to the captions, unless I am misinterpreting something. There is also a marked decrease in quality when noise is added.
- There are minor errors in the bibliography (e.g. PDEs should be capitalised in [30], the article A should not in [29], inconsistency after colons eg. between [32] and [27], in capitalisation between [42] and most other entries)

Decision letter (RSPA-2021-0916.R0)

25-Feb-2022

Dear Dr Sbalzarini

The Editor of Proceedings A has now received comments from referees on the above paper and would like you to revise it in accordance with their suggestions which can be found below (not including confidential reports to the Editor).

Please submit a copy of your revised paper within four weeks - if we do not hear from you within this time then it will be assumed that the paper has been withdrawn. In exceptional circumstances, extensions may be possible if agreed with the Editorial Office in advance.

Please note that it is the editorial policy of Proceedings A to offer authors one round of revision in which to address changes requested by referees. If the revisions are not considered satisfactory by the Editor, then the paper will be rejected, and not considered further for publication by the

journal. In the event that the author chooses not to address a referee's comments, and no scientific justification is included in their cover letter for this omission, it is at the discretion of the Editor whether to continue considering the manuscript.

To revise your manuscript, log into <http://mc.manuscriptcentral.com/prsa> and enter your Author Centre, where you will find your manuscript title listed under "Manuscripts with Decisions." Under "Actions," click on "Create a Revision." Your manuscript number has been appended to denote a revision.

You will be unable to make your revisions on the originally submitted version of the manuscript. Instead, revise your manuscript and upload a new version through your Author Centre.

When submitting your revised manuscript, you will be able to respond to the comments made by the referee(s) and upload a file "Response to Referees" in Step 1: "View and Respond to Decision Letter". Please provide a point-by-point response to the comments raised by the reviewers and the editor(s). A thorough response to these points will help us to assess your revision quickly. You can also upload a 'tracked changes' version either as part of the 'Response to reviews' or as a 'Main document'.

IMPORTANT: Your original files are available to you when you upload your revised manuscript. Please delete any unnecessary previous files before uploading your revised version.

When revising your paper please ensure that it remains under 28 pages long. In addition, any pages over 20 will be subject to a charge (£150 + VAT (where applicable) per page). Your paper has been ESTIMATED to be 22 pages.

Open Access

You are invited to opt for open access, our author pays publishing model. Payment of open access fees will enable your article to be made freely available via the Royal Society website as soon as it is ready for publication. For more information about open access please visit <https://royalsociety.org/journals/authors/open-access/>. The open access fee for this journal is £1700/\$2380/€2040 per article. VAT will be charged where applicable. Please note that if the corresponding author is at an institution that is part of a Read and Publishing deal you are required to select this option. See <https://royalsociety.org/journals/librarians/purchasing/read-and-publish/read-publish-agreements/> for further details.

Once again, thank you for submitting your manuscript to Proc. R. Soc. A and I look forward to receiving your revision. If you have any questions at all, please do not hesitate to get in touch.

Yours sincerely
Raminder Shergill
proceedingsa@royalsociety.org

Reviewer(s)' Comments to Author:

Referee: 1

Comments to the Author(s)

The paper by Suryanarayana et al presents an approach for learning interpretable and robust PDE models from data. The topic is surely relevant; the method consists of an adaptation of regularised regression techniques (with L_0 regularisation) combined with a stability-based model selection criterion (as opposed to say a more traditional cross-validation error measure). The paper is well written and thoroughly evaluated; my impression is that, while it may not be

radically innovative, it has certainly value and would be a good addition to the literature. My main comments are as follows:

- my understanding is that the IHT-d solver is a new heuristic, so it is not clear to me whether it inherits the guarantees cited by the authors for other algorithms. Which in turn may impact the "standard" choices for the parameters. Some clarification here would be useful.
- The PDE-STRIDE results are evaluated thoroughly on three simulated setups (perhaps two would suffice) plus one real example. All comparisons are however between different regularisers, I would also like to see in practice (maybe on only one example) practical advantages vis a vis an existing software tool.
- I would not go as far as calling PDE-STRIDE parameter free, I understand the qualifications but it seems a stretch.
- is the assumption of uncorrelated noise realistic? I would imagine that in real data there would be both spatial and temporal correlations in the noise. How would PDE stride work if the noise was e.g. a Gaussian process with a certain (perhaps small) spatial/ temporal correlation length? An experiment would be ideal but at least a discussion is needed.
- the real data application is somewhat underdeveloped. I would like to know whether the identified PDE makes biological sense. Additionally, shouldn't there be some conservation laws in place? How could these be enforced? Please also write out the final PDE system.

Referee: 2

Comments to the Author(s)

In this paper the authors propose a methodology for inferring PDEs from noisy measurement data. They provide what I believe is a valuable contribution to an active research field by combining stability selection and sparsity-promoting regression techniques. The paper contains some comparisons to other algorithms and experiments which suggest that this method can be usefully applied to real-world problems such as in the PAR example. All in all a solid research project that I can only recommend be published in this journal.

The reason for my recommending a major revision is that the RS requires all authors to make their supporting data, code and materials available. While the authors here provide a GitHub repository with some code, which includes the main algorithm, the only experiments they provide code for are on the Burgers equation (unless I am missing something). The code is generally understandable, although the second half of the iterative solver file could benefit from more functional encapsulation.

Furthermore the first page of Section 3b) appears empty on my computer. Judging from the first sentence on page 12 there is some content missing.

Minor comments:

- Fig. 1: The main font (Comic Sans?) is not very legible, and smaller labels are too thin. I would consider scaling up everything and removing the black border.
- L8. Remove comma after "models". As it stands it sounds like predictive models have to be derived from first principles and symmetry arguments, which clashes with the remaining paragraph.
- The last paragraph of 2b is a bit unclear on the approximate debiasing. The upper bound should ideally be specified in the main text as otherwise it is difficult to follow the reasoning without looking at the SI.
- In the PAR example the translation from model components (P, P2A etc.) to mechanistic components should be made more clear.
- In Figures S6 and S7 it seems strange that all parameters end up with a very high importance measure. Neither figure suggests that perfect recovery is the case, contrary to the captions, unless I am misinterpreting something. There is also a marked decrease in quality when noise is added.

- There are minor errors in the bibliography (e.g. PDEs should be capitalised in [30], the article A should not in [29], inconsistency after colons eg. between [32] and [27], in capitalisation between [42] and most other entries)

Author's Response to Decision Letter for (RSPA-2021-0916.R0)

See Appendix A.

RSPA-2021-0916.R1 (Revision)

Review form: Referee 1

Is the manuscript an original and important contribution to its field?

Good

Is the paper of sufficient general interest?

Good

Is the overall quality of the paper suitable?

Good

Can the paper be shortened without overall detriment to the main message?

Yes

Do you think some of the material would be more appropriate as an electronic appendix?

No

Do you have any ethical concerns with this paper?

No

Recommendation?

Accept as is

Comments to the Author(s)

The authors did a good job of addressing my previous concerns, I have no remaining issues.

Review form: Referee 2

Is the manuscript an original and important contribution to its field?

Good

Is the paper of sufficient general interest?

Good

Is the overall quality of the paper suitable?

Good

Can the paper be shortened without overall detriment to the main message?

Yes

Do you think some of the material would be more appropriate as an electronic appendix?

No

Do you have any ethical concerns with this paper?

No

Recommendation?

Accept with minor revision (please list in comments)

Comments to the Author(s)

The authors have responded to most of my and my fellow reviewer's comments, and the paper to me seems scientifically sound. However, I am still hesitant to recommend unconditional acceptance as my only major point of criticism has not been addressed: the absence of code for 3 out of 4 experiments, without which the paper would of course be much less interesting. I appreciate that this is a rather unrewarding task and I do not want to throw a spanner in the works of publishing this paper, but in the interest of reproducibility and given the complexity of the algorithm this does seem important.

I apologise if my original comment may have suggested a lack of code quality as my main gripe rather than a lack of code. Having a documented software package will surely encourage more people to use this method, but for the purpose of publication the issue was rather the missing experiments. I will leave the decision to the editorial policy of this journal.

Decision letter (RSPA-2021-0916.R1)

29-Apr-2022

Dear Professor Sbalzarini,

On behalf of the Editor, I am pleased to inform you that your Manuscript RSPA-2021-0916.R1 entitled "Stability selection enables robust learning of partial differential equations from limited noisy data" has been accepted for publication subject to minor revisions in Proceedings A. Please find the referees' comments below.

The reviewer(s) have recommended publication, but also suggest some minor revisions to your manuscript. Therefore, I invite you to respond to the reviewer(s)' comments and revise your manuscript. Please note that we have a strict upper limit of 28 pages for each paper. Please endeavour to incorporate any revisions while keeping the paper within journal limits. Please note that page charges are made on all papers longer than 20 pages. If you cannot pay these charges you must reduce your paper to 20 pages before submitting your revision. Your paper has been ESTIMATED to be 28 pages. We cannot proceed with typesetting your paper without your agreement to meet page charges in full should the paper exceed 20 pages when typeset. If you have any questions, please do get in touch.

It is a condition of publication that you submit the revised version of your manuscript within 7 days. If you do not think you will be able to meet this date please let me know in advance of the due date.

To revise your manuscript, log into <https://mc.manuscriptcentral.com/prsa> and enter your Author Centre, where you will find your manuscript title listed under "Manuscripts with Decisions." Under "Actions," click on "Create a Revision." Your manuscript number has been appended to denote a revision.

You will be unable to make your revisions on the originally submitted version of the manuscript. Instead, revise your manuscript and upload a new version through your Author Centre.

When submitting your revised manuscript, you will be able to respond to the comments made by the referee(s) and upload a file "Response to Referees" in Step 1: "View and Respond to Decision Letter". Please provide a point-by-point response to the comments raised by the reviewers and the editor(s). A thorough response to these points will help us to assess your revision quickly. You can also upload a 'tracked changes' version either as part of the 'Response to reviews' or as a 'Main document'.

IMPORTANT: Your original files are available to you when you upload your revised manuscript. Please delete any redundant files before completing the submission process.

When uploading your revised files, please make sure that you include the following as we cannot proceed without these:

- 1) A text file of the manuscript (doc, txt, rtf or tex), including the references, tables (including captions) and figure captions. Please remove any tracked changes from the text before submission. PDF files are not an accepted format for the "Main Document".
- 2) A separate electronic file of each figure (tif, eps or print-quality pdf preferred). The format should be produced directly from original creation package, or original software format.
- 3) Electronic Supplementary Material (ESM): all supplementary materials accompanying an accepted article will be treated as in their final form. Note that the Royal Society will not edit or typeset supplementary material and it will be hosted as provided. Please ensure that the supplementary material includes the paper details where possible (authors, article title, journal name). Supplementary files will be published alongside the paper on the journal website and posted on the online figshare repository (<https://figshare.com>). The heading and legend provided for each supplementary file during the submission process will be used to create the figshare page, so please ensure these are accurate and informative so that your files can be found in searches. Files on figshare will be made available approximately one week before the accompanying article so that the supplementary material can be attributed a unique DOI. Alternatively you may upload a zip folder containing all source files for your manuscript as described above with a PDF as your "Main Document". This should be the full paper as it appears when compiled from the individual files supplied in the zip folder.

Article Funder

Please ensure you fill in the Article Funder question on page 2 to ensure the correct data is collected for FundRef (<http://www.crossref.org/fundref/>).

Media summary

Please ensure you include a short non-technical summary (up to 100 words) of the key findings/importance of your paper. This will be used for to promote your work and marketing purposes (e.g. press releases). The summary should be prepared using the following guidelines:

*Write simple English: this is intended for the general public. Please explain any essential technical terms in a short and simple manner.

*Describe (a) the study (b) its key findings and (c) its implications.

*State why this work is newsworthy, be concise and do not overstate (true 'breakthroughs' are a rarity).

*Ensure that you include valid contact details for the lead author (institutional address, email address, telephone number).

Cover images

We welcome submissions of images for possible use on the cover of Proceedings A. Images should be square in dimension and please ensure that you obtain all relevant copyright permissions before submitting the image to us. If you would like to submit an image for consideration please send your image to proceedingsa@royalsociety.org

Open Access

You are invited to opt for open access, our author pays publishing model. Payment of open access fees will enable your article to be made freely available via the Royal Society website as soon as it is ready for publication. For more information about open access please visit <https://royalsociety.org/journals/authors/open-access/>. The open access fee for this journal is £1700/\$2380/€2040 per article. VAT will be charged where applicable. Please note that if the corresponding author is at an institution that is part of a Read and Publishing deal you are required to select this option. See <https://royalsociety.org/journals/librarians/purchasing/read-and-publish/read-publish-agreements/> for further details.

Once again, thank you for submitting your manuscript to Proceedings A and I look forward to receiving your revision. If you have any questions at all, please do not hesitate to get in touch.

Best wishes

Raminder Shergill

proceedingsa@royalsociety.org

Proceedings A

Reviewer(s)' Comments to Author:

Referee: 2

Comments to the Author(s)

The authors have responded to most of my and my fellow reviewer's comments, and the paper to me seems scientifically sound. However, I am still hesitant to recommend unconditional acceptance as my only major point of criticism has not been addressed: the absence of code for 3 out of 4 experiments, without which the paper would of course be much less interesting. I appreciate that this is a rather unrewarding task and I do not want to throw a spanner in the works of publishing this paper, but in the interest of reproducibility and given the complexity of the algorithm this does seem important.

I apologise if my original comment may have suggested a lack of code quality as my main gripe rather than a lack of code. Having a documented software package will surely encourage more people to use this method, but for the purpose of publication the issue was rather the missing experiments. I will leave the decision to the editorial policy of this journal.

Referee: 1

Comments to the Author(s)

The authors did a good job of addressing my previous concerns, I have no remaining issues.

Decision letter (RSPA-2021-0916.R2)

12-May-2022

Dear Professor Sbalzarini

I am pleased to inform you that your manuscript entitled "Stability selection enables robust learning of differential equations from limited noisy data" has been accepted in its final form for publication in Proceedings A.

Our Production Office will be in contact with you in due course. You can expect to receive a proof of your article soon. Please contact the office to let us know if you are likely to be away from e-mail in the near future. If you do not notify us and comments are not received within 5 days of sending the proof, we may publish the paper as it stands.

As a reminder, you have provided the following 'Data accessibility statement' (if applicable). Please remember to make any data sets live prior to publication, and update any links as needed when you receive a proof to check. It is good practice to also add data sets to your reference list. *Statement (if applicable):* The git repository for the codes and data can be found at <https://git.mpi-cbg.de/mosaic/software/machine-learning/pde-stride>.

Open access

You are invited to opt for open access, our author pays publishing model. Payment of open access fees will enable your article to be made freely available via the Royal Society website as soon as it is ready for publication. For more information about open access please visit <https://royalsociety.org/journals/authors/which-journal/open-access/>. The open access fee for this journal is £1700/\$2380/€2040 per article. VAT will be charged where applicable.

Note that if you have opted for open access then payment will be required before the article is published – payment instructions will follow shortly.

If you wish to opt for open access then please inform the editorial office (proceedingsa@royalsociety.org) as soon as possible.

Your article has been estimated as being 23 pages long. Our Production Office will inform you of the exact length at the proof stage.

Proceedings A levies charges for articles which exceed 20 printed pages. (based upon approximately 540 words or 2 figures per page). Articles exceeding this limit will incur page charges of £150 per page or part page, plus VAT (where applicable).

Under the terms of our licence to publish you may post the author generated postprint (ie. your accepted version not the final typeset version) of your manuscript at any time and this can be made freely available. Postprints can be deposited on a personal or institutional website, or a recognised server/repository. Please note however, that the reporting of postprints is subject to a

media embargo, and that the status the manuscript should be made clear. Upon publication of the definitive version on the publisher's site, full details and a link should be added.

You can cite the article in advance of publication using its DOI. The DOI will take the form: 10.1098/rspa.XXXX.YYYY, where XXXX and YYYY are the last 8 digits of your manuscript number (eg. if your manuscript number is RSPA-2017-1234 the DOI would be 10.1098/rspa.2017.1234).

For tips on promoting your accepted paper see our blog post:
<https://royalsociety.org/blog/2020/07/promoting-your-latest-paper-and-tracking-your-results/>

On behalf of the Editor of Proceedings A, we look forward to your continued contributions to the Journal.

Sincerely,
Raminder Shergill
proceedingsa@royalsociety.org

Appendix A

Referee: 1

The paper by Suryanarayana et al presents an approach for learning interpretable and robust PDE models from data. The topic is surely relevant; the method consists of an adaptation of regularised regression techniques (with L_0 regularisation) combined with a stability-based model selection criterion (as opposed to say a more traditional cross-validation error measure). The paper is well written and thoroughly evaluated; my impression is that, while it may not be radically innovative, it certainly has value and would be a good addition to the literature.

>> We thank the referee for recognizing the importance of our research and for acknowledging our efforts.

My main comments are as follows:

- my understanding is that the IHT-d solver is a new heuristic, so it is not clear to me whether it inherits the guarantees cited by the authors for other algorithms. Which in turn may impact the "standard" choices for the parameters. Some clarification here would be useful.

>> IHT-d is an extension of the popular Iterative Hard Thresholding (IHT) algorithm. IHT-d uses an additional approximate de-biasing step to increase the convergence and efficiency of the IHT algorithm while inheriting the guarantees of the original algorithm. We now mention this in the manuscript.

- The PDE-STRIDE results are evaluated thoroughly on three simulated setups (perhaps two would suffice) plus one real example. All comparisons are however between different regularisers, I would also like to see in practice (maybe on only one example) practical advantages vis a vis an existing software tool.

>> Our purpose was to compare different regularizers within one and the same framework on benchmark data with known ground truth. STRidge solves the same non-convex problem as the IHT-d algorithm. STRidge is the state-of-the-art PDE inference algorithm to date. To be objective in our comparison and for the sake of completeness, we now provide additional results using Orthogonal Matching Pursuit (OMP), a greedy algorithm that has been developed to find a sparse solution vector of an under-determined linear system of equations. We use the popular and widely used *scikit-learn* implementation of OMP in order to compare with an existing software tool, as suggested by the referee, and we report the results for comparison (last column) in the updated Figure 3 of the main text.

- I would not go as far as calling the PDE-STRIDE parameter free, I understand the qualifications but it seems a stretch.

>> We agree with the remark, and decided to rephrase these statements. However, PDE-STRIDE does enable us to recover the correct support with minimal parameter tuning for varying designs of the dictionaries using standard default values of the parameters.

- is the assumption of uncorrelated noise realistic? I would imagine that in real data there would be both spatial and temporal correlations in the noise. How would PDE stride work if the noise was e.g. a Gaussian process with a certain (perhaps small) spatial/ temporal correlation length? An experiment would be ideal but at least a discussion is needed.

>> This is a great remark. There is no reason to believe that our IHT-d algorithm will differentiate large or even small but finite correlations in the data from the actual dynamical signal. However, short length or time scale correlations are usually filtered by the prior de-noising step based on SVD. Long-range correlations in space and time, however, usually lead to filling the recovered support with miscellaneous terms of non-dynamical origin in order to account for these correlations. We now discuss this aspect in the revised conclusions section.

- the real data application is somewhat underdeveloped. I would like to know whether the identified PDE makes biological sense. Additionally, shouldn't there be some conservation laws in place? How could these be enforced? Please also write out the final PDE system.

>> The premise of the research question in the paper is being able to learn an *interpretable* model from a library of non-linear and differential operators. For the PAR problem, the inferred model is good at semi-quantitatively predicting the spatiotemporally varying protein concentration data, while also revealing the underlying antagonistic interactions between the proteins. However, the model doesn't necessarily conserve mass or obey flux balance. But these symmetries and conservation laws could be included into the learning problem if one wants to. We have addressed this issue in our subsequent work (new Ref. [79]) by extending PDE-STRIDE to include conservation laws and symmetries. In the revised manuscript, we now discuss this in the conclusions section. Also, as suggested by the referee, we have made the learned equations explicit and detailed their biological interpretation.

Referee: 2

In this paper the authors propose a methodology for inferring PDEs from noisy measurement data. They provide what I believe is a valuable contribution to an active research field by combining stability selection and sparsity-promoting regression techniques. The paper contains some comparisons to other algorithms and experiments which suggest that this method can be usefully applied to real-world problems such as in the PAR example. All in all a solid research project that I can only recommend be published in this journal.

>> We thank the reviewer for recommending our work to the journal.

The reason for my recommending a major revision is that the RS requires all authors to make their supporting data, code and materials available. While the authors here provide a GitHub repository with some code, which includes the main algorithm, the only experiments they provide code for are on the Burgers equation (unless I am missing something). The code is generally understandable, although the second half of the iterative solver file could benefit from more functional encapsulation.

>> We have made our github page and the code more legible with more detailed functional encapsulation. Currently, a software engineer in our group is working on turning the code into an encapsulated and well-documented Python package that could be installed by the Python Package Manager PIP.

Furthermore the first page of Section 3b) appears empty on my computer. Judging from the first sentence on page 12 there is some content missing.

>> Section 3b contains an important plot indeed. We are sorry it was not visible on your computer. We double-checked the revised PDF on several different viewers and computer systems and hope that it works for you as well now. If not, please reach out to the journal office who have the original source files.

Minor comments:

- Fig. 1: The main font (Comic Sans?) is not very legible, and smaller labels are too thin. I would consider scaling up everything and removing the black border.

>> We thank the reviewer for this suggestion. We have scaled the labels and made them more legible.

- L8. Remove comma after "models". As it stands it sounds like predictive models have to be derived from first principles and symmetry arguments, which clashes with the remaining paragraph.

>> We have reformulated and shortened the first sentence.

- The last paragraph of 2b is a bit unclear on the approximate debiasing. The upper bound should ideally be specified in the main text as otherwise it is difficult to follow the reasoning without looking at the SI.

>> We thank the reviewer for the helpful suggestion. We have revised the text accordingly.

- In the PAR example the translation from model components (P, P2A etc.) to mechanistic components should be made more clear.

>> In the revised draft, we have explicitly written the inferred model and also briefly describe the mechanistic interpretation associated with each component of the inferred model.

- In Figures S6 and S7 it seems strange that all parameters end up with a very high importance measure. Neither figure suggests that perfect recovery is the case, contrary to the captions, unless I am misinterpreting something. There is also a marked decrease in quality when noise is added.

>> The reviewer is correct in pointing out the seemingly conflicting results in the plot and its caption. The plot demonstrates that contrary to stability selection + IHT-d (main text Fig. 7), stability selection + STRidge is unable to recover the true model from data at high noise levels. This failure to recover the right model is attributed to the fact that many of the model components (blue shaded lines) are mixed with the noise components (black dashed lines). We have made corrections to the captions in order to better explain the plots. Similar changes to Figures S4 and S5 were also made to reflect the right interpretation from the plots.

- There are minor errors in the bibliography (e.g. PDEs should be capitalised in [30], the article A should not in [29], inconsistency after colons eg. between [32] and [27], in capitalisation between [42] and most other entries)

>> We have corrected the references for capitalization and other typos. Thank you very much for reporting them.